# Complex Principal Component Analysis of Mass Balance Change on Qinghai-Tibet Plateau

Jingang Zhan [1], Hongling Shi [1], Yong Wang [1*], Yixin Yao[1,2]

1State Key Laboratory of Geodesy and Earth's Dynamics, Institute of Geodesy and Geophysics, Chinese Academy of Sciences, Wuhan 430077, China

2University of Chinese Academy of Sciences, Beijing 100049, China

*Correspondence to*: Yong Wang (ywang@whigg.ac.cn)

**Abstract.** This paper reveals how climate changes affect spatial mass balance change across the Qinghai-Tibet Plateau. Such change is obtained using 153 monthly solutions of temporal gravity data from the Gravity Recovery and Climate Experiment satellite. Spatial mode, spatial phase distribution and principal components of the change are derived using complex principal component analysis. Time evolution of the major components is examined by wavelet analysis. Complex principal component analysis (particularly phase distribution) shows the trajectory of each factor that affects mass balance in the region, and the wavelet analysis shows time-frequency correlation between mass balance change and various atmospheric circulations. The first spatial mode indicates that mass change in the eastern Himalayas, Karakoram Pamirs and northwestern India was most sensitive to first principal component variation, which was responsible for 54.02% of that change. Correlation analysis shows that the first principal component is related to the Indian monsoon and the correlation coefficient is 0.828. The second spatial mode indicates that mass change on the eastern Qinghai plateau, eastern Himalayas-Qiangtang Plateau-Pamirs and northwestern India was most sensitive to variation of the second major factor, responsible for 16.38%. Correlation analysis also indicates that the second major component is associated with El Niño, the correlation coefficient is 0.302, almost twice as large as the 95% confidence level of 0.167. The third spatial mode shows that mass change on the western and northwestern QTP was most sensitive to climate change of its third major component, responsible for 5.64% of mass balance change. The third component may be associated with climate change from the westerlies and La Niña, because the third component and El Niño have similar signals of 6.5-yr period and opposite phase.

## 1 Introduction

The continuous rise of global sea levels presents a great challenge to the living environment of mankind. For example, storm tides strike coastal areas more frequently and flooding damage is increasing. The erosion of coasts and coastal lowlands causes beaches to recede. Water in coastal regions becomes polluted and farmlands are under threats to sanitation. Seawater absorbs heat and expands, causing global sea levels to rise (Willis, 2003; Antonov et al., 2005). Moreover, the rise of temperature accelerates the melt speed of polar ice caps and glaciers on land, with part of the meltwater directly (meltwater of polar ice caps) or indirectly (meltwater of glaciers) entering the sea through runoff, which can also lead to rising sea levels (Nguyen and Herring, 2005, Anny and Frédérique, 2011; Shi et al., 2011; Church et al., 2013). Furthermore, the melting of glaciers accelerates the loss of freshwater resources by which humans live. All these are results of global climate change.

As the cryosphere of the so-called "Third Pole," the Qinghai-Tibet Plateau (QTP) contains numerous glaciers and lakes water resources. Covering an area of 47,000 km$^2$, these glaciers are the headstreams of many famous Asian rivers. The plateau is famous for its altitude and vast territory, with a complicated developing environment for glaciers and a changeable climate. For example, the southern and southeast parts of the plateau are under the influence of the Indian and East Asian monsoon circulations, which bring abundant summer rain. The western part, where the Pamirs are located, is under the influence of westerlies that produce dry and rainless areas. However, the interior of the QTP is less influenced by aforementioned circulations and is dominated more by continental climatic conditions (Yao et al., 2012; Yi and Sun, 2014). The difference between the results of those authors and that of Yi et al.(2014) is that the latter believed that the Indian monsoon is much stronger than the westerlies and it can therefore also influence precipitation in the Pamirs in winter and summer. Compared with the findings of Yao et al., Yi and Sun neglected the influence of the East Asian monsoon. However, we hold that the developing environment of glaciers on the QTP is more complicated (as shown in Figure 1), because in recent years the El Niño phenomenon has become frequent and is gradually strengthening. Thus, we have enough evidence to believe that this phenomenon will influence the development of glaciers on the plateau.

A glacier is the most sensitive and direct information carrier of climate change. Their melting process records the most direct and detailed dynamic change information of local or even global climate. Glaciologists and meteorologists reproduce ancient climates and the environment by analyzing data of samples taken from glaciers in plateau areas.

They then study the response relationships between glaciers and ancient climate change on long time scales and forecast likely future climate change (Thompson et al., 2006; Yao and Yu, 2007; Yao et al., 2012). However, for plateaus with sparse populations, it is obviously unrealistic to obtain glacier time sequences with high spatial resolution.

The development of space geodetic technology, especially that of earth observation from space, provides researchers with highly precise and continuous earth observation data in terms of glacier mass change and water storage variation in untraveled regions. With these data, unprecedented research achievements have been made in evaluating mass balance in polar and Asian alpine regions (Chen et al., 2009; Matsuo and Heki, 2010; Chen et al., 2011; Gardelle et al., 2012; Jacob et al., 2012; Matsuo and Heki 2012; Yao et al., 2012; Gardelle et al., 2013; Gardner

et al., 2013; Yi and Sun, 2014; Xiang et al., 2016). In the application of Gravity Recovery and Climate Experiment (GRACE) observation data, their methods are generally similar. After subtracting signals of the glacial isostatic adjustment (GIA) model and terrestrial water storage model from the GRACE data, residual gravity change can be fully attributed to changes in glaciers. However, there is a lack of necessary analysis of the dissimilarity of spatial variation and its causes.

The change of mass balance in the cryosphere is the result of interactions between glaciers and atmosphere at different spatial and temporal scales. To study the time-varying spatial change of mass balance on the QTP, principal component analysis (PCA) is a useful method (Fenogliomarc, 2000; Wang et al., 2000). The greatest advantage of PCA is that it can describe complicated changes of initial datasets with fewer variables. However, traditional PCA can detect only a standing wave, not advancing waves, because of a lack of corresponding phase information. To overcome

this disadvantage, Wallace and Dickinson (2010) developed the complex principal component analysis in the frequency domain (FDPC) method. This performs principal component analysis by calculating vectors of complex features of a relative spectrum matrix. FDPC is the most common method to study spatiotemporal transmission characteristics. However, if climate change fluctuates over irregular time intervals and the energy of its principal component is distributed in multi-frequency bands, the spatial change image of every frequency spectrum must be

analyzed. In such a case, it is inconvenient to use FDPC. Compared with FDPC, complex principal component analysis (CPCA) in the time domain is attractive (Horel, 1984). The CPCA method transforms original data and its Hilbert transform into a complex time sequence and conducts principal component analysis by calculating the covariance or complex characteristics vector of the cross-correlation matrix. CPCA is an FDPC method for a full-frequency band.

When datasets only have a single frequency, CPCA is equivalent to FDPC. Therefore, CPCA can be used to effectively detect transmitting characteristics, especially when the variance of the principal component is distributed across many frequency bands.

In this paper, the 153 approximately monthly gravity solutions from GRACE Release 05 data are used to reproduce the spatial change of mass balance on the QTP. Then, the main components and corresponding spatial modes and time variation of the mass balance in this area are studied using the complex principal component analysis technique. The period of each principal component and its time evolution are also examined using the wavelet amplitude-period Spectrum Analysis in order to explore possible reasons for the spatial difference of mass balance over the QTP. This is very helpful to understand the response of mass balance to climate change in this region, and is very important to assess the potential impacts of glacier melt on water resources, ecology and environmental disasters.

**2 Data**

The variation of earth's gravity field reflects the redistribution of mass inside the earth. Over a short time (compared with geologic time), it can be regarded as mass transfer of the earth's surface and shallow fluid. GRACE, which was jointly developed by the U.S. and Germany, has been successfully operating for over 10 years. Its monthly gravity solutions have been able to reflect changes of 1-mm geoid fluctuation at 300-km spatial scale and can be used to monitor gravity field variations caused by changes in hydrology and the cryosphere, earthquakes and glacial isostatic adjustment (Ramillien et al., 2006; Chen et al., 2007; Chen et al., 2008; Velicogna, 2009; Rignot et al., 2011).

The time-varying gravity model used in this paper is the Release-05 (RL05) solutions provided by the Center for Space Research (CSR), University of Texas at Austin. The 153 approximately monthly GRACE gravity solutions cover the period January 2003 through September 2015 (~12 solutions are missing), each of which consist of normalized spherical harmonic (SH) coefficients, to degree and order 60. The main improvements in the new products are the mean gravity model and corrections of various new tide models. Some processing algorithms and parameters have also been improved, regarding alignments between the star camera data rate, accelerometer, and K-band system (Bettadpur, 2012). Compared with previous data, the RL05 gravity solutions substantially reduced the stripe noise and has the ability to monitor 1 mm geoid undulation at the spatial scale of 300 km (Bettadpur et al,2015; Save et al.,2016). However, at high degrees and orders, GRACE spherical harmonics are contaminated by noise, including

longitudinal stripes, and filtering is still needed. In our study, the smoothness priors method (Tarvainen et al., 2002;

Zhan. et al., 2015) was used to remove noise in the spatial domain. Compared with the Gaussian filter, Correlated-

Error filter and the combined filter (Gaussian with 300 km smoothing + Correlated-Error), the smoothness priors

method has advantages of less reduction in signal amplitude at high latitude, preservation of greater detail for short-

wavelength components in the result and less signal distortion at low latitude. Moreover, grid statistical results of the

filtered field show that the result of smoothness priors method is the most similar to the actual in the minimum,

maximum and the RMS values of the original field (Zhan. et al., 2015).

## 3 Method

### 3.1 Equivalent Water Height

According to Wahr et al., (1998), surface mass change can be expressed in the form of surface equivalent water

height (EWH) as

$$\Delta\sigma(\theta, \lambda) = \frac{a\rho_e}{3\rho_w} \sum_{n=0}^{\infty} \frac{2n+1}{1+k_n} \sum_{m=0}^{n} \left\{ \left[ \tilde{c}_n^m \cos(m\lambda) + \tilde{s}_n^m \sin(m\lambda) \right] \tilde{P}_n^m(\cos\theta) \right\} \tag{1}$$

where $\rho_e$ is average density of the earth, $a$ is the equatorial radius, and $\rho_w$ is water density. Parameter $\lambda$ is

longitude, $\theta$ is colatitude, and $\tilde{P}_n^m(\cos\theta)$ is the nth-degree and mth-order fully normalized Legendre function.

Parameter $k_n$ is the load Love number. $\tilde{c}_n^m$ and $\tilde{s}_n^m$ are normalized SH coefficients.

### 3.2 CPCA

Principal component analysis (PCA) was first formulated in statistics by Pearson (1901), Hotelling (1932)

further developed PCA to its present stage. Since then, the utility of PCA has been rediscovered in many diverse

scientific fields, and it now goes under many names, such as singular value decomposition (SVD) (Golub et al.,

1996; Mandel, 1982) and empirical orthogonal function (EOF) analysis (Lagerloef et al.,1988; Kaihatu et al.,1998;

Zhang et al.,2004). Eigenvector analysis and characteristic vector analysis are often used in the physical sciences

and other fields.

PCA (Abdi et al.,2010; Helena et al., 2000; Wang et al.,2000) is a multivariate technique that analyzes a data

table in which observations are described by several inter-correlated quantitative dependent variables. Its goal is to extract the important information from the table, represent it as a set of new orthogonal variables called principal

components, and display patterns of similarity of the observations and variables as points in maps. Mathematically, PCA depends upon the eigen-decomposition of positive semi-definite matrices and SVD of rectangular matrices.

Compared with PCA, the CPCA method (Horel, 1984) introduces phase information and was shown to be a useful method for identifying traveling and standing waves (Pfeffer et al., 2010; Kichikawa et al., 2015). CPCA transforms original data and its Hilbert transform into a complex time sequence and conducts principal component analysis by

calculating the covariance or complex characteristics vector of the cross-correlation matrix.

For the CPCA, a complex observation sequence should first be constructed, which is different from the PCA. For a time varying observation vector $u_j(t)$, its Fourier expansion is:

$$u_j(\text{t}) = \sum_{\omega} \left[ a_j(\omega)\cos(\omega t) + b_j(\omega)\sin(\omega t) \right]. \tag{2}$$

In this expansion, $j$ stands for the location of the observation point, $t$ is the observation time, and $\omega$ is the Fourier

frequency. The constructed complex observation vector $U_j(t)$ can be expressed as

$$U_j(t) = \sum_{\omega} c_j(\omega)e^{-i\omega t} \tag{3}$$

Here, $c_j(\omega) = a_j(\omega) + ib_j(\omega), i = \sqrt{-1}$. According to the definition of $c_j(\omega)$, Eq. (3) can be expanded as

$$U_j(t) = \sum_{\omega} \left[ a_j(\omega)\cos(\omega t) + b_j(\omega)\sin(\omega t) \right] + i \left[ b_j(\omega)\cos(\omega t) - a_j(\omega)\sin(\omega t) \right]$$
$$= u_j(t) + i\text{v}_j(t) \tag{4}$$

The real part of Eq. (4) is the original observation sequence and the imaginary part is the Hilbert transform of the real

part, which does not change the amplitude of each component of $u_j(t)$. However, the phase of each spectral component is advanced by $\pi/2$.

The traditional PCA is principal component analysis of the real observation vector, whereas CPCA analysis is such analysis of the constructed complex vector. After normalization of the complex observation vectors, that is the average value is subtracted from the complex observation vector of each observation point, and then divided by the

standard deviation the complex correlation matrix of the observation point can be expressed as:

$$
\begin{bmatrix}
r_{11} & r_{12} & \cdots & r_{1n} \\
r_{21} & r_{22} & \cdots & r_{2n} \\
\vdots & \cdots & \cdots & \vdots \\
r_{n1} & r_{n2} & \cdots & r_{nn}
\end{bmatrix} . \tag{5}
$$

Here $r_{jk}$ represents the multiple correlation coefficients between the $j$th and $k$th observation points. CPCA compresses

information using the least complex eigenvector $e_{jn}$ of correlation matrix (Eq. 5) and the complex principal

component $p_n(t)$, because the correlation matrix (5) is a Hermitian matrix including n real eigenvalues $\lambda$ .

$\lambda_j / \sum\limits_{i=1}^{n} \lambda_i$    denotes the contribution percentage of the $j$th principal component.

Observation vector $U_j(t)$ can be expressed as the sum of N principal components,

$$
U_j(t) = \sum_{n=1}^{N} e_{jn}^{*} p_n(t), \tag{6}
$$

where * stands for the complex conjugate, and both complex principal components and complex eigenvectors are

orthogonal. The $n$th complex eigenvector element $e_{jn}$ can be expressed as

$e_{jn} = \left[ U_j(t) * p_n(t) \right]_t = s_{jn} e^{i\theta_{jn}}$ . \hfill (7)

Where, $e_{jn}$ indicates the multiple correlation relationship between the jth time sequence and nth principal component.

$s_{jn}$ and $\theta_{jn}$ are respectively correlative order of magnitude and phase. $\left[ \cdots \right]_t$ signifies the average of times. The

time sequence elements of principal components can be expressed as the functional form of amplitude $T_n$ and phase

$\Phi_n$ .

$P_n(t) = T_n(t) e^{i\Phi_n(t)}$ \hfill (8)

### 3.3 Wavelet Amplitude-period Spectrum Analysis

Mass balance on the QTP is under the influence of climate change, and exhibits unsteady quasi-periodic change.

After obtaining the temporal change series of principal components in the area, the time-varying changes of the periods

and amplitude (energy) should be analyzed. Here, we use the wavelet amplitude-period spectrum (Liu, 1999; Liu and

Hsu, 2012; Zhan et al., 2003) to analyze its time-frequency information, and choose Morlet wavelet (Morlet et al., 2012) as the basic wavelet. The wavelet amplitude-period spectrum reflects the time-varying amplitude and period of each periodic term (or standardized periodic term). This means that in this spectrum, the location of extreme points corresponds to the instant period of a periodic signal (or quasi periodic term) at that moment, whereas the extreme point value corresponds to the instantaneous amplitude of a certain period signal at that moment. The wavelet amplitude-period spectrum of a time sequence $f(t)$ is defined as

$$W_\psi f(a,b) = \frac{1}{aC_\psi} \int_{-\infty}^{\infty} f(t)\psi\left(\frac{t-b}{a}\right) dt, \quad a,b \in R, \quad a \neq 0, \tag{9}$$

where $\psi(t) = e^{\frac{-t^2}{2\delta^2}} \cos(2\pi\omega_0 t), \quad \delta, \omega_0 \in R, 2\pi\delta\omega_0 \gg 1. \quad C_\psi = \int_{-\infty}^{\infty} \psi(t)\cos(2\pi\omega_0 t)dt$

Here, the kernel function $\psi(t)$ is the real part of the Morlet wavelet, $\delta$ is a constant, and $\omega_0$ is the frequency parameter, $C_\psi$ is a constant, $a$ and $b$ are scale factors of period and time, respectively.

**4 Mass change and its CPCA analysis**

A regional $1° \times 1°$ gridded (24° - 45°N , 70°-105°E) surface mass change field (in units of equivalent water height) was calculated from each GRACE spherical harmonic solutions following Equations (1). Then, we filtered each surface mass change field using the smoothness priors method (Tarvainen et al., 2002; Zhan et al., 2015) and interpolated missing data using a spline function at each grid point. Finally, GRACE mass rate was estimated at each grid point using the least squares to fit a linear trend, plus annual and semiannual sinusoids to GRACE-derived mass change time series. As the fitting results, the amplitude values of annual and semiannual terms are constants, so the calculated trend values contain the contributions from the annual and semiannual trend. It's worth noting that, $1° \times 1°$ gridded data used here does not represent that the resolution of GRACE data has been improved. The resolution of the calculated data depends on the degree of the RL05 solutions and the GRACE RL05 solutions are limited by the band-limited nature of GRACE orbit configuration (inclination, altitude, and separation of the twin satellites), with an approximate resolution of around 300 km near the equator (Chen et al., 2016). People can also get relevant information from NASA websites (https://grace.jpl.nasa.gov/data/get-data/jpl_global_mascons/). One can also calculate smaller grid data using those solutions, the smaller calculated grid data does not mean that there are more short-wavelength

 Figure 2 shows the trend of mass balance on the QTP during the period 2003-2015. Figure 2 shows that this mass balance on the QTP has two major change characteristics, namely, a large negative signal with mass decrease around the southern edge of the plateau (Himalayas and its southern region) and a positive signal with mass increase over inland areas of the plateau. However, in the Pamirs region, mass variation had no obvious trend. Here, we analyzed mass variation in the area during 2003–2015 using CPCA in order to analysis the reasons for mass change. Before the CPCA analysis, data of mass change were filtered, and missing data were interpolated at each grid point. Table 1 shows corresponding eigenvalues of the first five principal components and their contribution percentages to mass change in the area. We took the example of the first three principal components for explanation and description. According to table 1, the result from CPCA of mass variation over the QTP shows that the eigenvalues of the first, second and third principal components are respectively 82.6516, 25.0562 and 8.6290, and their contribution percentages are respectively 54.02%, 16.38% and 5.64%, which can explain 76.04% of the variation of mass balance in the area.

Figure 3a shows the first spatial mode and its spatial phase distribution (arrows) from the CPCA analysis of the mass balance change in the area. According to the figure, the first spatial mode shows change characteristics of three areas: two negative signals of the eastern Himalayas to the Hengduan Mountains (AB area) and the Pamirs to the Karakorum Mountains (D area), and a positive signal in the northwestern India (H area). The direction of the arrows indicates the sequence of mass change and the arrow size the change rate of mass. It is obvious from the phase information that the first spatial mode mainly reflects the character of mass change, which is from south to north.

Figure 3b and 3c depict the temporal evolution of the first principal component and its wavelet amplitude-period spectrum analysis results, respectively. It is seen in figure 3c that the periodic component that affected the first spatial mode is mainly annual periodic signal, whose period and amplitude are relatively stable. According to the result of the time-sequence wavelet amplitude-period spectrum, the period components of the first spatial mode time sequence are very simple, which are single annual-period signals featuring steady periods. The result of its wavelet amplitude-period spectrum is the same as the result of the wavelet amplitude-period spectrum of the Indian monsoon indices time sequence (Figure 3d).

We examined possible relationships between the first principal component and the Indian monsoon indices by calculating their lag correlation coefficient and corresponding 95% confidence level based on Monte Carlo Hypothesis testing (table 2). The lag correlation coefficient of the first principal component with the Indian monsoon indices is 0.828, much larger than the 95% confidence level of 0.223, and change of the first principal component lags that of India monsoon indices by one month. Obviously, there is significant correlation between them. From the phase information of mass variation and the correlation analysis, it can be inferred that the first spatial mode in the area is strongly controlled by the Indian monsoon, revealing the influence of that monsoon on rainfall in various areas and its spatial evolution. A branch of the monsoon enters the QTP via the AB area and proceeds northward over the Tanggula Mountains with gradually declining energy. It is then blocked by the Qilian Mountains and turns westward, forming a circulation. Another branch proceeds northward and enters the Qiangtang Plateau from the middle and western part of the Himalayas, and is obstructed by the Kunlun and Altun mountains. It then progresses westward into the Pamirs. According to Table 1, the influence of the Indian monsoon accounts for 54.02% of mass balance change on the QTP. According to the time sequence of the spatial mode (Figure 3b), the Indian monsoon has been weakening since 2009, and the change of that monsoon is the main reason for mass balance change in the area.

Figure 4a shows the second spatial mode and its phase information. From this, it is seen that this mode is mainly manifested as three mass change zones of southeast–northwest orientation: a positive signal in the southern Karakorum–northwestern India, two negative signals in the AB area–Qiangtang Plateau (E area)–Karakorum, and the southern Qilian Mountains. Red arrows in the figure show phase information of the second spatial mode, whose direction change is relatively disordered. They mainly enter the inland plateau from the southeast and affect its mass balance change.

Figure 4b and 4c respectively show the temporal evolution of the second principal component in the area and its wavelet amplitude-period spectrum analysis. From the result of the wavelet amplitude-period spectrum of its time series, we see that the periodic component of the second principal component is relatively complicated. It mainly contains a semiannual cycle signal, annual cycle signal, 2-4-year and 6.5-year cycle signals. The semiannual, annual and 6.5-year cycle signals have the strongest energy. Energy in the 2-4-year cycle signal is relatively weak, and their energy are all unstable. Comparing with the wavelet amplitude-period spectrum (Figure 4d) of El Niño evolution in corresponding periods, it was found that both have 6.5-year and annual cycle signals with consistent phase position.

Similarly, we also examined possible relationships between the second principal component and El Niño by calculating their correlation coefficient and corresponding 95% confidence level based on Monte Carlo hypothesis testing (table 2). Their correlation coefficient is 0.302, nearly twice as large as the 95% confidence level of 0.167. Change of the second principal component lags that of El Niño by one month. The test result shows a strong correlation between them. According to the spatial phase information and wavelet amplitude-period spectrum, the data suggest that the second spatial mode in the area is mainly affected by climate change related to the East Asian monsoon and El Niño. Its influence is largely divided into two branches. One enters the Qinghai Plateau through the Sichuan basin, and the other enters the Qiangtang Plateau through the eastern Himalayas and extends to the northwest of the plateau until reaching the Karakorum mountain region and then turns south.

Figure 5a portrays the third spatial mode and its spatial phase distribution information (arrows). The figure shows that the third mode is mainly revealed by the features of two regions, a positive signal in the middle-western area (west of 90°E) and a negative signal in the region of Linzhi (A area). Mass change in other regions is in a weak state of balance. The red arrow in the figure shows phase distribution information of the third spatial mode; its direction shows that the mass change had an obvious west-to-east configuration. This indicates that the factors behind the change of this mode came from the western part.

Figure 5b and c shows the time change series of the third spatial mode and its wavelet transform spectrum in the area. From results of this wavelet transform spectrum, the cycle components of this mode mainly contain semiannual, annual, 2-4-year and 6.5-year cycle signals. In contrast with the results of the second main component, energy of the time series of the third spatial mode mainly concentrates in a 2-6.5-year periodic signal; the annual cycle signal is relatively weak. Except for the 6.5-year signal, energy of the cycle signals is not stable. The phase of the 6.5-year cycle signals in the second and third main components are opposite, which indicates that their driving mechanisms are opposite.

According to the spatial phase information, we conclude that the third spatial mode is mainly affected by the westerlies and La Niña phenomenon, whose influence can be divided into three branches. One branch moves to the north beyond the Karakorum Mountains and enters the Tarim Basin, and then reaches the eastern Qinghai Plateau. Another branch moves east beyond the western Himalayas and enters the Qiangtang Plateau. Then it meets the East Asian monsoon around 90°E and is obstructed. The third branch goes southward along the Himalayas and influences

northern India. The westerlies are weak in the south and strong in the north, so a clear northeast-southwest boundary

of force range (blue line in Fig. 5a) is formed in the inland part of the QTP.

## 5 Discussion

### 5.1 Mass Change in Inland QTP

In the inland part of the QTP, there are three obvious mass increase regions, the Qiangtang Plateau (E area),

middle and east of the Kunlun Mountains (F area), and Qinghai Plateau (G area). Their respective annual increases

were 4.5, 5.5 and 3.5 GT, much smaller than the 30 GT of Yi and Sun (2014). Many scholars have conducted related

research in an attempt to explain the reason behind mass balance change in the region.

Mass balance of the Inner Tibet Plateau (ITP) derived from GRACE data showed a positive rate that was

attributable to glacier mass gain, whereas those glaciers from other field-based studies showed an overall mass loss.

For example, Jacob et al. (2012) deduced glacier mass balance using GRACE data, finding a mass increase rate of 7

Gt yr$^{-1}$ in the E and F areas. However, according to onsite observation of more than 20 glaciers in QTP area (Yao et

al., 2012), glaciers in that area are shrinking dramatically. Their results indicate that the Himalayas have shown the

most extreme glacial shrinkage based on the reduction both of glacier length and area. The shrinkage is most

pronounced in the southeastern QTP, where the length decreased at a rate of 48.2m yr$^{-1}$ and the area declined at a rate

of 0.57% yr$^{-1}$ during the 1970s-2000s. The rate of glacial shrinkage decreased from the southeastern QTP to the interior.

Zhang et al. (2013) studied 53% of the total lake area on the plateau using ICESAT satellite data, finding a mass

increase rate of 4.95 Gt yr$^{-1}$. They suggested that the increased mass measured by GRACE was largely due to increased

water mass in lakes. If this rate holds true for all lakes, the total mass variance rate is +8.06 Gt yr$^{-1}$ according to the

area ratio. However, glacier melting into lakes, itself, should not increase the overall mass and may decrease the mass

because a portion of the meltwater would be lost through evaporation or discharged to rivers that leave the Tibet

Plateau.

Yi and Sun (2014) indicated a relatively large mass rate change in this area, and explained this change through

glacier change, lake water levels, geologic structural processes, and frozen soil. They stated that according to model

calculation, the change of inland water storage was −3.3 Gt yr$^{-1}$. The change of negative balance of weakening glacier

mass has been confirmed (Bolch et al., 2010, Bolch et al., 2012, Yao et al., 2012). According to the calculation and

estimation of Zhang et al. (2013), the increase of lake water is 8.1 Gt yr$^{-1}$, and the effect of tectonic movement (simple Bouguer correction) is 0–13 Gt yr$^{-1}$. The effect of other factors is close to zero. However, we still lack enough observation data of mass balance states in the interior part of the earth in the study region. Thus, the exact Bouguer equilibrium correction requires more scientific data for confirmation.

The effect of soil freezing on mass change in the inland plateau is weak, because the terrain there is flatter than at the plateau edge. The inland area contains numerous lakes and wetlands, which is conducive to the convergence of fluid. Moreover, when water melts and is lost from frozen soil, soil porosity definitely increases, which captures more water during rainy periods.

        Based on the results of our work, we tend to support the point that rainfall is the main reason for the mass increase

in the study region. Data indicate strong evidence that precipitation over the inland QTP during the past several decades has greatly increased (Yao et al., 2012; Global Precipitation Climatology Project or GPCP, www.esrl.noaa.gov/psd/data /gridded/data.gpcp.html ). On one hand, influenced by El Niño, moist air moves westward to the inland plateau through the eastern Himalayas and Qinghai, which brings rainfall to the inland area and causes rainfall accumulation in plateau lakes and wetland areas. On the other hand, the inland Plateau,

especially the western part of Qiangtang plateau and Kunlun mountains area, is also influenced by the westerlies and La Niña phenomenon (Figure 5a), which further create the meteorological conditions for rain and snow. Moreover, the increase of temperature (Qin et al., 2009) accelerates glacial melting in this area. This glacier meltwater enters lakes through runoff. It also explains why onsite observation data of glaciers indicate slight shrinkage, and GRACE observations indicate the reason for mass increase.

**5.2 Mass Change of Glaciers in Himalayas Region**

        The trend of mass balance change from GRACE data shows that the most negative signal is along the Himalayas and northwestern India. The mass reduction rate of glaciers in the entire Himalaya mountain region is 14 Gt yr$^{-1}$, and the mass loss of glaciers in the eastern Himalayas was the most dramatic, with the rate of -4.6 Gt yr$^{-1}$ in A area and -4.1 Gt yr$^{-1}$ in B area. The mass reduction rate in northwestern India (H area) was −13.6 Gt yr$^{-1}$, whereas Rodell et al.

(2009) and Yi et al. (2014) gave larger values of −17.7 Gt yr$^{-1}$ and −20.2 Gt yr$^{-1}$, respectively. The reason for this discrepancy is that Rodell et al. (2009) used the data of the RL04 version. Yi and Sun (2014) stated that the RL04 solutions tend to overestimate the glacier melt rate in the Himalayas by as much as 17%. The difference between our

results and those of Yi and Sun (2014) is because they used the mascon inverse method in a concise form. Moreover, the filtering method may somewhat attenuate the signal.

Yao et al. (2012), after investigating the glacial change over the past 30 years, reported that the Himalayas shows the most extreme glacial shrinkage based on the reduction both of glacier length and area, the shrinkage is most significant in the southeastern QTP (A area), where the length decreased at a rate of 48.2 m yr$^{-1}$ and the area was reduced at a rate of 0.57% yr$^{-1}$ during the 1970s-2000s, and the most negative mass balances occurred along the Himalayas, ranging from -1100 to -760mm yr$^{-1}$. This trend of mass change along the Himalayas is consistent with our
result. They attribute this change to the weakened Indian monsoon towards the interior of the plateau.

Thakuri et al. (2014) examined glacier changes on the south slope of Mt. Everest from 1962 to 2011 (400 km$^2$) using optical satellite imagery and concluded that the observed glaciers shrinkage, upward shift of snowline altitudes, and the negative mass balance (Nuimura et al., 2012) is not only due to warming temperatures, but also as a result of weakened Asian monsoons registered over the last few decades. Bolch et al. (2011) examined the mass change of
glaciers on Mt. Everest, Nepal using stereo Corona spy imagery (years 1962 and 1970), aerial images, and recent high resolution satellite data (Cartosat-1), founding that glaciers south of Mt. Everest had continuously lost mass from 1970 through 2007, with a possibly increasing rate in recent years. Wagnon et al. (2013) arrived at the same conclusion. They also indicated that glacier shrinkage south of Mt. Everest was less than that of others in the western and eastern Himalaya and southern and eastern Tibetan Plateau.

Recently, Salerno et al. (2015) analyzed the precipitation time series during 1994-2013 reconstructed from seven stations located between 2660 and 5600 m a.s.l. They found that precipitation even decreased 47% during the monsoon period and the snowfall decreased 10% in the last 20 years. Salerno et al. (2016) extended this analysis to even the first 1960s and for all regions used, as proxy of the precipitation trend, the surface area variation of glacial lakes. These authors inferred an increase in precipitation occurred until the mid-1990s followed by a decrease until recent
years in all Mt. Everest regions.

Studies using different types of data arrived to the same results: i.e. negative mass balances and weakened Indian monsoon along Himalayas. Our results support this conclusion, the results of CPCA analysis indicate that mass change on the Himalayas and its southern portion are associated with the Indian monsoon climate, and the intensity of this monsoon is weakening. This result is also consistent with the conclusions of Wu (2005). A weakened Indian monsoon

brings less humid air to the study region, causing interannual rainfall decreases, (Thakuri et al., 2014; Salerno et al.,

2015, 2016). The GPCP rainfall data confirms this conclusion. The eastern Himalayas are also affected by El Niño

(figure 4a) and East Asian monsoons, and no evidence supporting the role of westerlies (figure 5a) in driving local

climate and glacier changes. Glaciers in this area are of a marine type, whose mass has large inputs and outputs and is

strongly affected by changes of marine climate. The weakened Indian monsoon, strengthening El Niño and westerlies,

combined with the huge topographic landform, exert climatic controls on the distribution of existing glaciers along all

Himalayas regions and bring more less precipitation to there.

**5.3 Effect of Circulation in QTP area**

Archer et al. (2004) indicated that the western Hindu-Kush Karakoram is largely exposed to the arrival of

westerly midlatitude perturbations bringing precipitation during winter and early spring, whereas the eastern Himalaya

is dominated by summer monsoon precipitation (Syed et al. 2006; Yadav et al. 2012). There is little difference between

their results and ours. The results of CPCA indicate that the eastern Himalaya is under the influence of weakened

Indian monsoon and El Niño, while the Hindu-Kush Karakoram area is under the influence of a weaken Indian

monsoon, westerlies and La Niña.

Thompson et al. (2000) examined the variability of the South Asian monsoon by analyzing ice core records of

Dasuopu glacier on the QTP, finding evidence of drought conditions and a weakened monsoon from 1780 to 1810.

Interestingly, according to historical recorders, at least 600,000 people died in 1972 in just one region of northern

India from an epic drought associated with this event. The onset of this event in the Dasuopu cores is concurrent with

a very strong El Niño–Southern Oscillation (ENSO) of 1790-1793, which was followed by a moderate ENSO event

of 1794-1797 as documented. These data suggest an association between ENSO and weakened Asian monsoon.

Studies have suggested that Arctic amplification may impact mid-latitude weather patterns and extremes (Francis

et al., 2012; Screen et al., 2013), and mid-latitude westerlies may drive climate variation and glacier variability in

monsoon affected areas of High Asia (Thomas et al., 2014). On large spatial scales, climate change over the QTP may

also be teleconnected with hemispheric or global atmospheric circulations including North Atlantic Oscillation (NAO)

and ENSO (Wang et al., 2003). Some literature suggests that ENSO influences climate over the southern QTP through

a link with the Indian monsoon (Xu et al., 2010; Xu et al., 2011). The NAO is associated with climate fluctuations

over the northern QTP through modulation of the westerlies (Wang et al., 2003; Xu et al., 2010), which is similar to climate change from the westerlies and La Niña in the third principal component.

Through observation of the glacial change over the past 30 years on the QTP area, Yao et al. (2012) discovered that glacier recession in the Himalayas was the most dramatic, followed by the inland plateau. Glaciers in the Pamirs had weak balance changes, and some of the glaciers in the eastern Pamirs Plateau are still expanding. Yao et al. (2012) believed that the main reason for this phenomenon was the change of climates with different circulations, which includes effects of the weakened Indian monsoon in the Himalayas and rainfall decreases, also includes effects of the strengthening of the westerlies in the Pamirs and its eastern portion and rainfall increases. While in the inland plateau, the influences of these two circulations are limited. The two atmospheric circulation patterns, combined with the huge topographic landform, exert climatic controls on the distribution of existing glaciers. The East Asian monsoon only affects glaciers on the eastern margin, such as the Mingya Gongga and those in the eastern Qilian Mountains. They believed that the interior of the QTP is dominated more by continental climatic conditions, and the sparse glacier distribution and higher ELAs in the continental-climate-dominated interior are consequences of a limited water-vapor source from both those air masses. They divided glaciers of the Tibet Plateau into seven regions and categorized them into three climatic transects: transect 1, southwest-northeast oriented (middle Himalaya-Qiangtang Plateau-eastern Qinghai Plateau), with the weakened Indian monsoon influence northward; transect 2, southeast-northwest oriented (eastern Himalayas-Qiangtang Plateau-Pamirs), with the weakened Indian monsoon toward the interior and strengthening westerlies toward the northwest; and transect 3, along the Himalayas, with stronger monsoon influence in the east and weaker monsoon influence in the west.

To some extent, we support this type of classification. From results of the CPCA, the first spatial mode clearly shows that the mass balance of the Himalayas-Pamirs-northwestern India (transect 3) was the most sensitive to climate change associated with the Indian monsoon, whereas the impact of that change on mass balance of the inland plateau was not very sensitive. The third spatial mode shows that mass balance of the northwest plateau, including all the Kunlun mountains (not only the Pamirs and its eastern portion), is also affected by climate change from the westerlies and La Niña. Another difference between the results of Yao et al. (2012) and ours is that climate change from El Niño rather than the weakened Indian monsoon toward the interior affected mass balance along transect 2, because we found that the time evolution of the second principal component and El Niño index had a stronger time-frequency correlation.

Through harmonic analysis of the time series of mass changes in the study region, Yi and Sun (2014) found a 5 year periodic signal in the Pamirs and Karakorum regions. Then, they analyzed correlation between mass change,

precipitation, El Niño–Southern Oscillation (ENSO) and the Arctic Oscillation (AO), founding that the 5-year undulating signal of mass change is controlled by both the ENSO and AO.

Recently, Ke et al.(2017) examined area and thickness change of glaciers in the Dongkemadi (DKMD) region of the central QTP using Landsat images from 1976 to 2013 and satellite altimetry data from 2003–2008. They then analyzed relationships between glacier variation and local and macroscale climate factors based on various remote

sensing and reanalysis data. Their results suggest that glacier change in the DKMD region was dominated by the variation of mean annual temperature, and was influenced by the state of the NAO over the past 38 years. The mechanism linking climate variability over the central QTP and state of the NAO is most likely via changes in strength of the westerlies and Siberian High. In addition, ENSO may have been associated with extreme weather (snowstorms) in October 1986 and 2000 which might have led to substantial glacier expansion in the following years. It is noteworthy

that the DKMD is located on the eastern Qiangtang Plateau (the center of transect 2), where area mass balance change is the most sensitive to El Niño in our results.

Yao et al. (2012) considered the effect of the Indian monsoon and westerlies but ignored the phenomenon of El Niño, which was the second major component (16.38%) in the study region. Yi and Sun (2014) also noted that the five-year periodic signal in the Pamirs region is related to ENSO, but ignored the effect of La Niña because they did

not distinguish the phase information. According to the CPCA, we believe that the mass change in the QTP area is mainly controlled by the Indian monsoon and westerlies, and the influence of El Niño and La Niña on the inland of plateau and Karakorum area cannot be ignored. The Indian monsoon mainly affects mass balance change on southern and southwestern QTP, whereas El Niño mainly modifies that change over the eastern Himalayans, Qiangtang Plateau, Pamirs and eastern Qinghai Plateau area. Mass balance over the western and northwestern QTP is mainly affected by

the westerlies and La Niña.

**6 Conclusions**

During 2003-2015, mass change on the Tibetan Plateau and surroundings varied systematically from region to region. Specifically, the Himalayas region (along Himalayas) had the greatest negative mass balance with mass

decrease at a rate of -14 Gt yr$^{-1}$ and the continental interior of the plateau had a positive signal with mass increase at a rate of 13.5 Gt yr$^{-1}$, whereas the Pamirs had a weak negative mass balance. The main cause of the systematic mass change was the variation of rainfall, which mainly results from changes in four different atmospheric circulation patterns over the QTP and surroundings, i.e., the weakening Indian monsoon, strengthened westerlies, El Niño, and La Niña. Their contributions can explain approximately 76.04% of mass change on the QTP.

Change of the Indian monsoon was the most important effect on mass balance variation over the QTP. The lag correlation coefficient of the first principal component with the Indian monsoon indices is 0.828, much larger than the 95% confidence level of 0.223, and the change of the first principal component lags that of the India monsoon indices by one month. Mass balance variation over the eastern Himalayan Mountains, Karakoram, Pamirs and northwestern India was the most sensitive to change of the Indian monsoon, and was responsible for 54.02% of that change. The weakened Indian monsoon, combined with the huge topographic landform, exerted climatic controls on the distribution of existing glaciers in these regions and caused less precipitation there.

Because El Niño is strengthening, it has recently become the second major effect on mass balance change of QTP, and was responsible for 16.38% of that change. Their lag correlation coefficient is 0.302, almost twice the 95% confidence level of 0.167, and change of the second principal component lags that of El Niño by one month. Mass balance over the eastern Himalayas, Qiangtang Plateau, Pamirs and eastern Qinghai Plateau areas were the most sensitive to El Niño variation. Further research is needed to better understand the physical mechanisms linking El Niño and mass balance.

The third principal component was climate change of the westerlies and La Niña. Mass balance on the western and northwestern QTP were the most sensitive to climate change from the westerlies and La Niña, which represented 5.64% of mass balance change. The strengthening westerlies and La Niña climate phenomenon created meteorological conditions for rain and snow to those regions, and there is no evidence in our results to support the role of westerlies in driving glacier changes across the southeastern QTP.

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

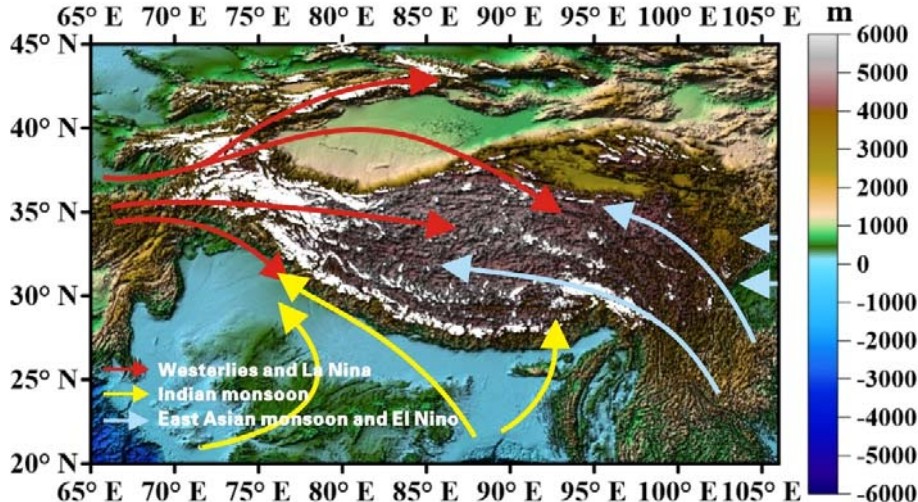

**Figure 1: Distribution of glaciers (white dots) and atmospheric circulation in and around Tibetan Plateau.**


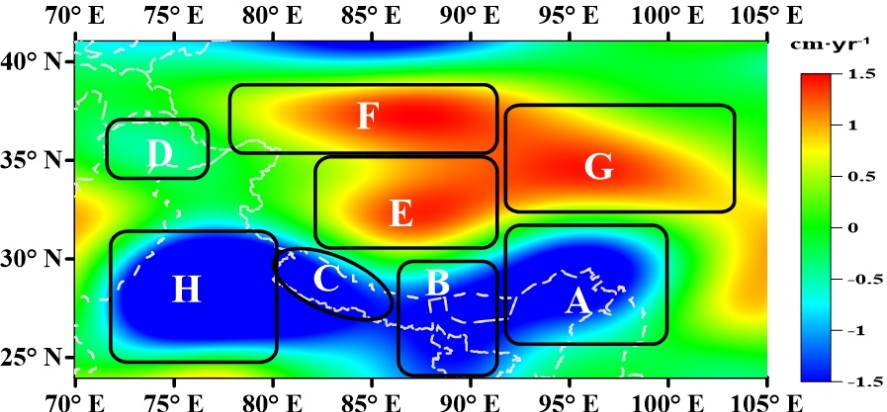

**Figure 2 Trend of mass balance in and around Tibetan Plateau. (A)Eastern Himalaya, (B) central Himalaya, (C) western Himalaya, (D) Pamirs, (E) Qiangtang Plateau, (F) Kunlun mountain, (G) Qinghai plateau, (H) northwestern India.**


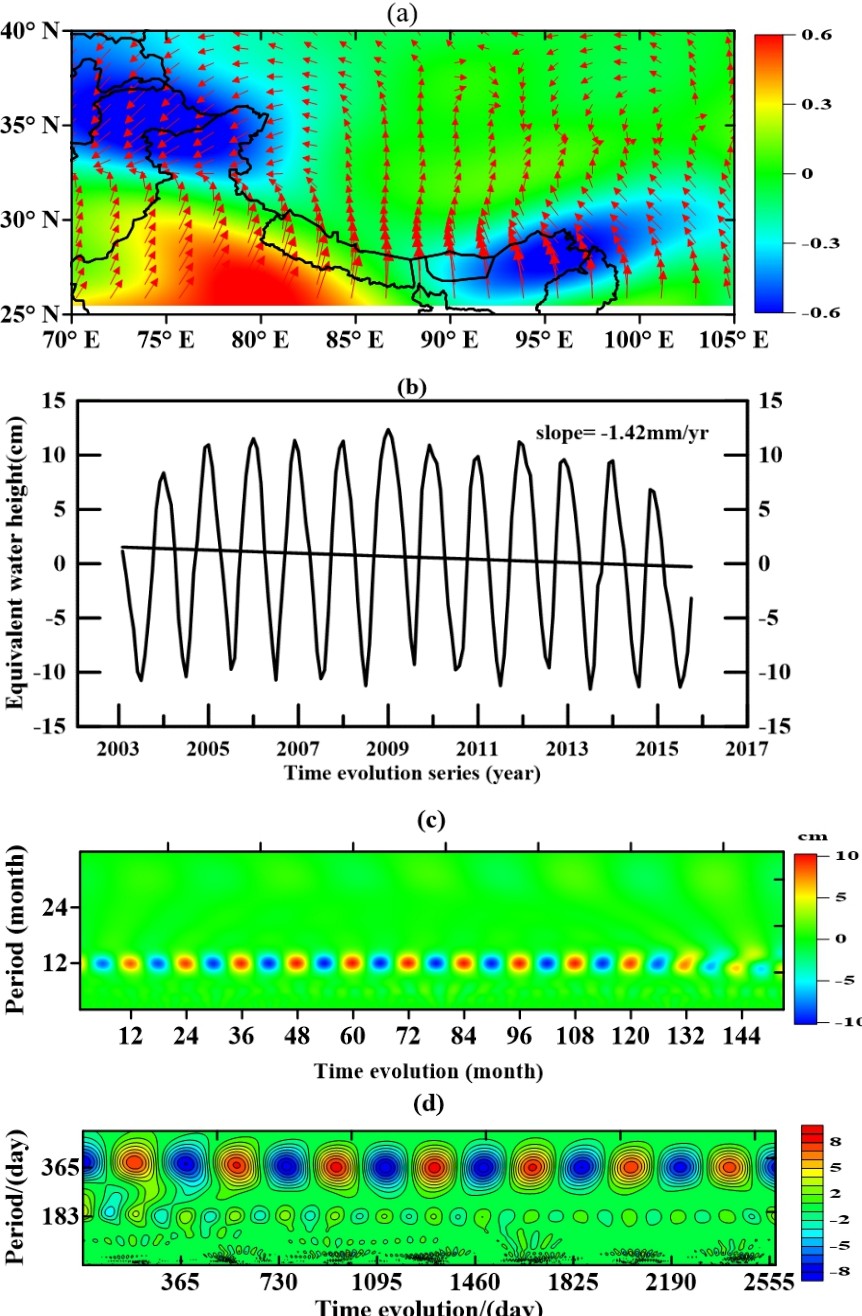

**Figure 3 First spatial mode and phase (red arrows) (a), temporal patterns of first principal component (b), and its wavelet amplitude-period spectrum (c) of mass balance change, as well as wavelet amplitude-period spectrum of Indian monsoon indices in period 2003–2009 (d)**



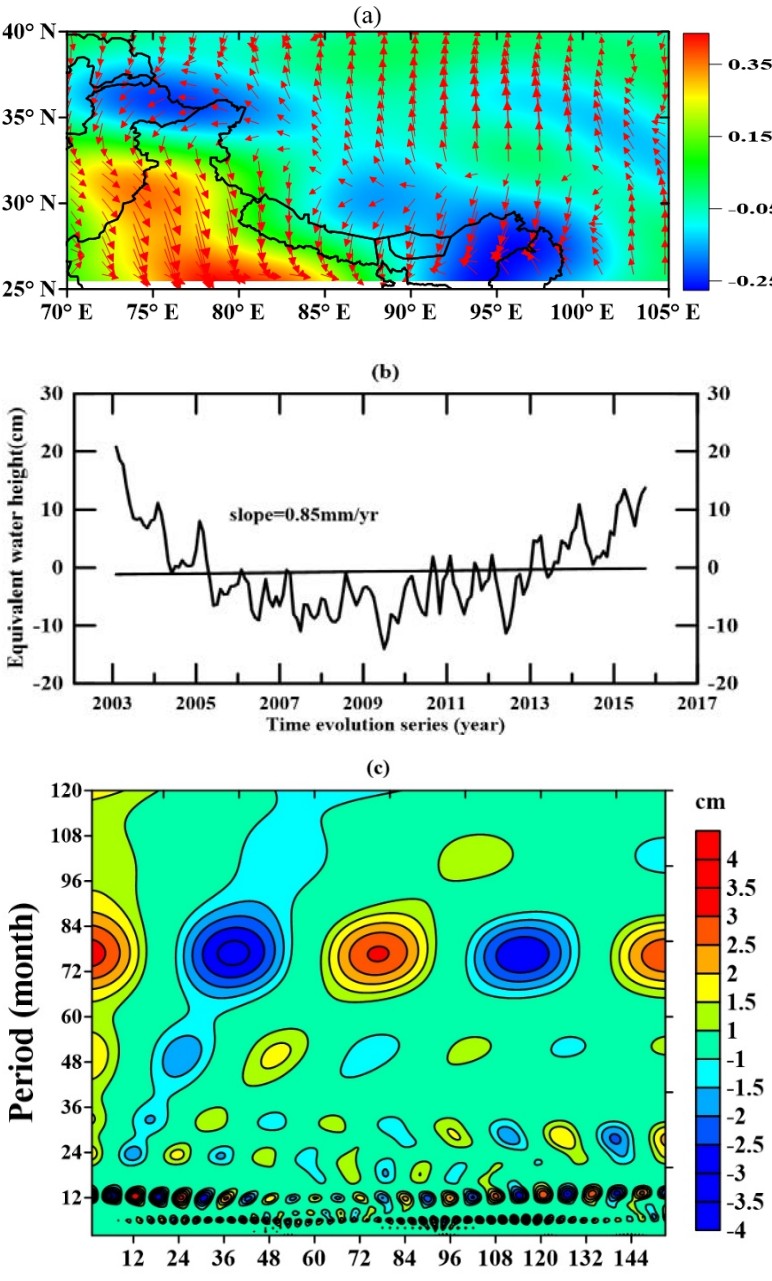

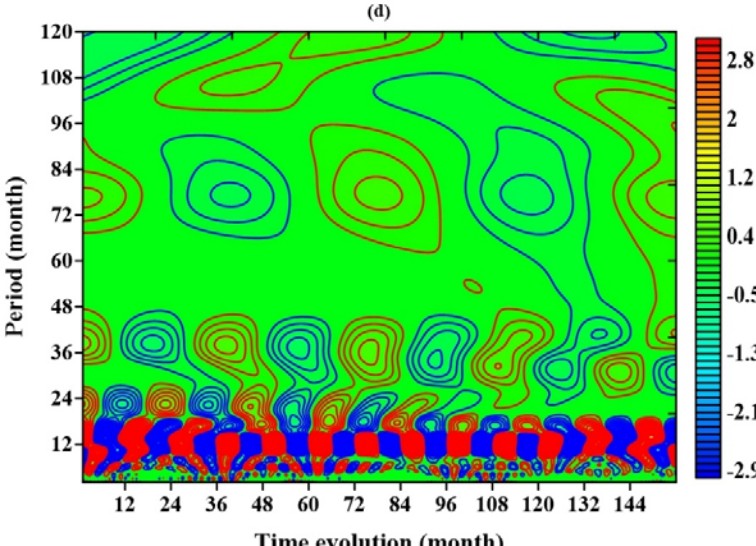

Figure 4 Second spatial mode and phase (red arrows) (a), temporal patterns of second principal component (b), and its wavelet amplitude-period spectrum (c) of mass balance change, as well as wavelet amplitude-period spectrum of El Niño in period 2003–2015 (d)

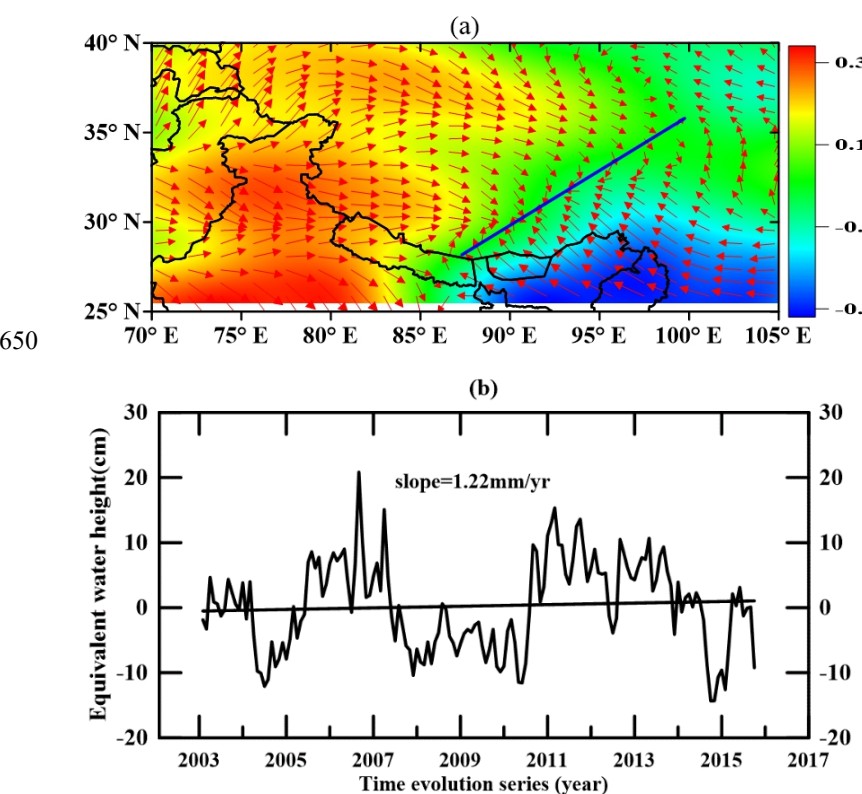

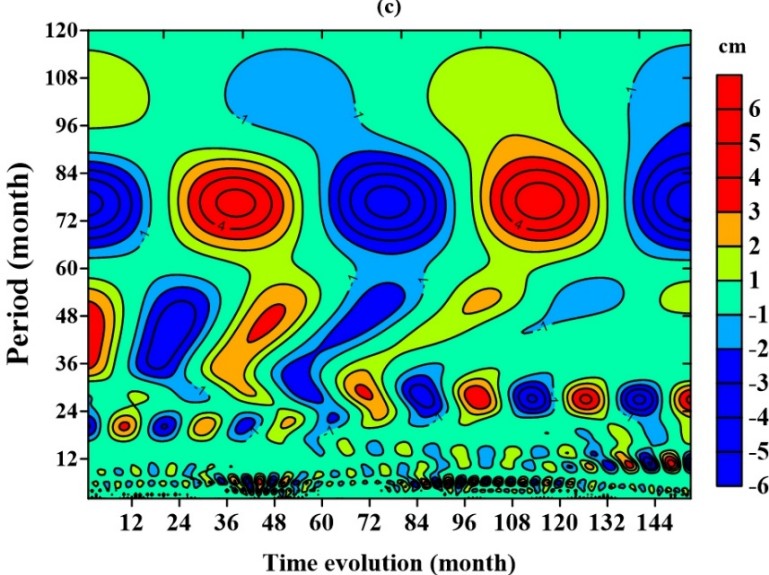

**Figure 5 Third spatial mode and phase (red arrows) (a), temporal patterns of third principal component (b), and its wavelet amplitude-period spectrum (c) of mass balance change.**

**Table 1 Eigenvalues and contribution percentages to mass change in CPCA analysis of Qinghai-Tibet Plateau**

| Number | Eigenvalues | As percentages | Cumul. percentages |
|---|---|---|---|
| 1 | 82.6516 | 54.02 | 54.02 |
| 2 | 25.0562 | 16.38 | 70.40 |
| 3 | 8.6290 | 5.64 | 76.04 |
| 4 | 7.3688 | 4.82 | 80.85 |
| 5 | 5.1715 | 3.38 | 84.23 |

**Table 2 Correlation analysis based on Monte Carlo hypothesis testing**

| | Time lag (month) | First principal component | Second principal component | 95% confidence level |
|---|---|---|---|---|
| India monsoon indices | 1 | 0.828 | - | 0.228 |
| El Niño | 1 | - | 0.302 | 0.167 |
