# Peer review of "Complex Principal Component Analysis of Mass Balance Change on Qinghai-Tibet Plateau"

_The Cryosphere, 2016_

## Short Comment (SC1) · 24 Nov 2016

This paper, among the others results, shows that glacial fluctuation of the Himalayas area was mainly influenced by the weakening Indian monsoon.

Personally, I support these findings, because, at more local scale (southern slopes of Mt. Everest, central Himalaya), we arrived to the same results.

Thakuri et al., 2014 examining glacier changes from 1962 to 2011 (400 km2) concluded that the observed glaciers shrinkage, upward shift of snowline altitudes (SLAs) and the negative mass balance (Bolch et al., 2011; Nuimura et al., 2012) is not only due to warming temperatures, but also as a result of weakening Asian monsoons registered over the last few decades. The registered losses could be mainly due to a minor accumulation. Wagnon et al. (2013), in the same region, arrived at the same conclusion.

[Figure]

Recently, Salerno et al., 2015, analyzing the precipitation time series reconstructed through land weather stations located at high elevation (5000 m a.s.l.) shown that in the last 20 years precipitation decreased of even 47% during the monsoon period!!! Salerno et al., 2016 extended this analysis even the first 1960s and for all region using, as proxy of the precipitation trend, the surface area variation of glacial lakes. These authors inferred an increase in precipitation occurred until the mid-1990s followed by a decrease until recent years in all Mt. Everest region.

Bolch, T., Pieczonka, T., and Benn, D. I.: Multi-decadal mass loss of glaciers in the Everest area (Nepal Himalaya) derived from stereo imagery, The Cryosphere, 5, 349-358.

Nuimura, T., Fujita, K., Yamaguchi, S., and Sharma, R. R.: Elevation changes of glaciers revealed by multitemporal digital elevation models calibrated by GPS survey in the Khumbu region, Nepal Himalaya, 1992-2008, J. Glaciol., 58, 648-656, 2012.

Salerno F., N. Guyennon, S. Thakuri, G. Viviano, E. Romano, E. Vuillermoz, P. Cristofanelli, P. Stocchi, G. Agrillo, Y. Ma, and G. Tartari, 2015. Weak precipitation, warm winters and springs impact glaciers of south slopes of Mt. Everest (central Himalaya) in the last 2 decades (1994–2013). The Cryosphere 9, 1229-1247.

Salerno, F., Thakuri, S., Guyennon, N., Viviano, G., Tartari, G., 2016. Glacier melting and precipitation trends detected by surface area changes in Himalayan ponds. The Cryosphere, 10 (4), 1433-1448.

Thakuri S., F Salerno, C Smiraglia, T Bolch, C D'Agata, G Viviano, Tartari G., 2014. Tracing glacier changes since the 1960s on the south slope of Mt. Everest (central Southern Himalaya) using optical satellite imagery. The Cryosphere 8 (4), 1297-1315.

Wagnon, P., Vincent, C., Arnaud, Y., Berthier, E., Vuillermoz, E., Gruber, S., Ménégoz, M., Gilbert, A., Dumont, M., Shea, J. M., Stumm, D., and Pokhrel, B. K.: Seasonal and annual mass balances of Mera and Pokalde glaciers (Nepal Himalaya) since 2007, The

Cryosphere, 7, 1769-1786.

---

## Author Comment (AC1) · 30 Nov 2016

Thank you very much for your comments, which will help us to enrich our manuscript. Previous studies had done a lot of interesting works in Himalayan glacier change using satellite imagery, multi-temporal RS-DEMs and GPS data, as well as temperature and precipitation information (Thakuri et al., 2014; Bolch et al., 2011; Nuimura et al., 2012; Wagnon et al.,2013; Salerno et al., 2015& 2016). Here, the GRACE data also indicates that the Himalayan glaciers show a negative mass balance and was mainly influenced by the weakening Indian monsoon. In addition, compared with the interior plateau, the central and eastern Himalaya are weakly influenced by the westerlies and La Niña-related climate, which also lowered the chance of precipitation in this region to some extent. Our result from GRACE has a good agreement with the previous studies from other different data and provides further evidence of the weakening Indian monsoon in

this area(Yao et al.,2012;Yi et al.,2014). Thanks again!

---

## Referee Comment (RC1) · P. Kishore (Referee) · 30 Dec 2016

Review on "Complex Principal Component Analysis of Mass Balance Change on Qinghai-Tibet Plateau" by Jingang Zhan et al.,

The above paper studies on GRACE mass balance wave characteristics using Complex Principal Component Analysis (CPCA) during the period from January 2003 to September 2015 (153 months). The authors are mentioned spatial mode characteristics using CPCA analysis, but did not mention any spatial characteristics in the abstract. The abstract is not conveying major outcome of the study. In my point of view, there exist major deficiencies in terms of analysis and discussion of the manuscript. I cannot recommend this manuscript for publication in the present form and author should undertake a major revision by performing in-depth analysis and interpreting the results.

[Figure]

There are major concerns which need to be clarified on data analysis and interpretation of the results. They are listed below.

Pg1 line 13: "The results show that the mass balance change on the plateu is influenced by various atmospheric circulation...", this is general statement but this statement is not concluded by his analysis and why authors are mentioned in abstract?

Pg1 line 15: "circulations and there are obvious systematic differences..." In this statement author should give clear on "obvious"?. This type of obvious signal datasets create a huge impact on CPCA analysis.

Pg1 line 15: "change "systemic" to "systematic"

line 103: "filtering is still necessary to suppress high-degree and order...", authors should mention what type of filter is used for smoothing. Is it s time domain or area smoothing are used for this analysis?

Line 105: "the smoothness priors method (Tarvainen et al., 2002) was used to remove stripe noise in GRACE data", In CPCA analysis works best when dominant variability contained within the data set is concentrated in a narrow frequency band. In GRACE RL05 data has some spikes over equator region in some months. Authors should carefully handle the data, especially removing the spikes at the same time without loosing the mass change signal.

line 145: "Wavelet amplitude...", authors did not explained all terms in the equation (9) and authors are not explained clearly the wavelet analysis methodology. The author should provide the recent literature regarding the wavelet analysis.

Line159: "Figure 2 shows the trend of mass balance...", authors showed the spatial trends but did not mentioned how they calculated the trends? In Figure 2 is not clear and no trend scale.

Generally CPCA analysis allows the efficient detection of propagating features, especially when the variance spread over a number of frequencies. Before applying

the CPCA analysis the date shouldn't be discontinuous, sudden transitions, and noisy spikes. In addition in this manuscript many conclusive statements, which are only speculations, and language corrections are required. In my view, there exist major deficiencies in terms of analysis, clarity and discussion of the present manuscript.

---

## Referee Comment (RC2) · Anonymous Referee #2 · 5 Feb 2017

Based on complex principal component analysis (CPCA) and wavelet amplitude-period spectrum, this paper analyzes the spatial characteristics of mass balance change on the Qinghai-Tibet Plateau and surrounding areas, using 153 monthly solutions of temporal gravity data from GRACE. It was concluded that the major influence on the change of mass balance on the Qinghai-Tibet Plateau was the weakening Indian monsoon. The second major factor was El Niño. And the third was the westerlies and La Niña. Though some results are different from some other researchers', it is interesting to present some new explanation based on GRACE data.

Convincing evidences are required to support the authors' view.

How was the mass balance obtained in Figure 2?

Why choose the smoothness priors method for filtering? The authors should explain

more.

How to ensure reliable results with CPCA?

Is it suitable to only analyze the first three major factors?

"However, we believe that geologic structural processes are slow." Could the authors quantitatively describe the impact on mass balance change?

The discussion and conclusions can be improved.

The manuscript should be carefully revised. Some errors and suggestions are listed as follows.

There are no legends in some figures, such as Figure 2, Figure 3, Figure 4a and Figure 5a.

Line 22: the westerlies and of La Niña?

Line 48: TBP should be explained when it first appears.

Line 85: Qinghai Tibet Plateau?

Line 118: the left side of Eq. (3)?

Line 168: The number in Table 1 should be same as the number in line 168 and in the paper.

Line 273: the bracket?

Line 310: 74%?

---

## Author Comment (AC2) · 1 Mar 2017

**Responses to the comments of P. Kishore**

2017-2-27

| The comments from referee 1 | Responses | Changes made in the manuscript |
|---|---|---|
| 1. The authors are mentioned spatial mode characteristics using CPCA analysis, but did not mention any spatial characteristics in the abstract. The abstract is not conveying major outcome of the study. | Accepted | We revised the Abstract to convey the major outcomes of our study based on the suggestion.
**Revised on Pg1 lines 8–24:**
 'This paper reveals how climate changes affect spatial mass balance change across the Qinghai-Tibet Plateau. Such change is obtained using 153 monthly solutions of temporal gravity data from the Gravity Recovery and Climate Experiment satellite. Spatial mode, spatial phase distribution and principal components of the change are derived using complex principal component analysis. Time evolution of the major components is examined by wavelet analysis. Complex principal component analysis (particularly phase distribution) shows the trajectory of each factor that affects mass balance in the region, and the wavelet analysis shows time-frequency correlation between mass balance change and various atmospheric circulations. The first spatial mode indicates that mass change in the eastern Himalayas, Karakoram Pamirs and northwestern India was most sensitive to first principal component variation, which was responsible for 54.02% of that change. Correlation analysis shows that the first principal component is related to the Indian monsoon and the correlation coefficient is 0.828. The second spatial mode indicates that mass change on the eastern Qinghai plateau, eastern Himalayas-Qiangtang Plateau-Pamirs and northwestern India was most sensitive to variation of the second major factor, responsible for 16.38%. Correlation analysis also indicates that the second major component is associated with El Niño, the correlation coefficient is 0.302, almost twice as large as the 95% confidence level of 0.167. The third spatial mode shows that mass change on the western and northwestern QTP was most sensitive to climate change of its third major component, responsible for 5.64% of mass balance change. The third component may be associated with climate change from the westerlies and La Niña, because the third component and El Niño have similar signals of 6.5-yr period and opposite phase'. |

| 2. In my point of view, there exist major deficiencies in terms of analysis and discussion of the manuscript. I cannot recommend this manuscript for publication in the present form and author should undertake a major revision by performing in-depth analysis and interpreting the results. | Accepted | Some contents were supplemented to improve the analysis and discussion as the suggestion. |
|---|---|---|

Some contents were supplemented to improve the analysis and discussion as the suggestion.

**Pg9 lines 210-214:**

[revised manuscript text omitted]
 circulation ...", this is general statement but this statement is not concluded by his analysis and why authors are mentioned in abstract? | Accepted | This paper mainly reveals how the climate affect mass balance change over the Qinghai-Tibet Plateau by complex principal component analysis.

Here, we just want to stress the point that the four climate changes are important in mass balance of that plateau, and were responsible for 76.04% of that change.

We revised the Abstract to convey the major outcomes of our study, based on the suggestions

**Pg1 lines 8-24:**

'This paper reveals how climate changes affect spatial mass balance change across the Qinghai-Tibet Plateau. Such change is obtained using 153 monthly solutions of temporal gravity data from the Gravity Recovery and Climate Experiment satellite. Spatial mode, spatial phase distribution and principal components of the change are derived using complex principal component analysis. Time evolution of the major components is examined by wavelet analysis. Complex principal component analysis (particularly phase distribution) shows the trajectory of each factor that affects mass balance in the region, and the wavelet analysis shows time-frequency correlation between mass balance change and various atmospheric circulations. The first spatial mode indicates that mass change in the eastern Himalayas, Karakoram Pamirs and northwestern India was most sensitive to first principal component variation, which was responsible for 54.02% of that change. Correlation analysis shows that the first principal component is related to the Indian monsoon and the correlation coefficient is 0.828. The second spatial mode indicates that mass change on the eastern Qinghai plateau, eastern Himalayas-Qiangtang Plateau-Pamirs and northwestern India was most sensitive to variation of the second major factor, responsible for 16.38%. Correlation analysis also indicates that the second major component is associated with El Niño, the correlation coefficient is 0.302, almost twice as large as the 95% confidence level of 0.167. The third spatial mode shows that mass change on the western and northwestern QTP was most |

| | | |
|---|---|---|
| | | sensitive to climate change of its third major component, responsible for 5.64% of mass balance change. The third component may be associated with climate change from the westerlies and La Niña, because the third component and El Niño have similar signals of 6.5-yr period and opposite phase'. |
| 4. Pg1 line 15: "circulations and there are obvious systematic differences..." In this statement author should give clear on "obvious"?. This type of obvious signal datasets create a huge impact on CPCA analysis. | Accepted | The word "obvious" here is to express that the various regions of mass change are affected by different atmospheric circulations, based on the first three components.

We revised the Abstract to convey the major outcomes of this study, based on the suggestions:

**Pg1 lines 8-24:**

'This paper reveals how climate changes affect spatial mass balance change across the Qinghai-Tibet Plateau. Such change is obtained using 153 monthly solutions of temporal gravity data from the Gravity Recovery and Climate Experiment satellite. Spatial mode, spatial phase distribution and principal components of the change are derived using complex principal component analysis. Time evolution of the major components is examined by wavelet analysis. Complex principal component analysis (particularly phase distribution) shows the trajectory of each factor that affects mass balance in the region, and the wavelet analysis shows time-frequency correlation between mass balance change and various atmospheric circulations. The first spatial mode indicates that mass change in the eastern Himalayas, Karakoram Pamirs and northwestern India was most sensitive to first principal component variation, which was responsible for 54.02% of that change. Correlation analysis shows that the first principal component is related to the Indian monsoon and the correlation coefficient is 0.828. The second spatial mode indicates that mass change on the eastern Qinghai plateau, eastern Himalayas-Qiangtang Plateau-Pamirs and northwestern India was most sensitive to variation of the second major factor, responsible for 16.38%. Correlation analysis also indicates that the second major component is associated with El Niño, the correlation coefficient is 0.302, almost twice as large as the 95% confidence level of 0.167. The third spatial mode shows that mass change on the western and northwestern QTP was most sensitive to climate change of its third major component, responsible for 5.64% of mass balance change. The third |

| | | component may be associated with climate change from the westerlies and La Niña, because the third component and El Niño have similar signals of 6.5-yr period and opposite phase.' |
|---|---|---|
| 5. Pg1 line 15: "change "systemic" to "systematic" | Accepted | We revised the Abstract to convey the major outcomes of this study, based on the suggestions.

**Pg1 lines 8-24.** |
| 6. line 103: "filtering is still necessary to suppress high-degree and order...", authors should mention what type of filter is used for smoothing. Is its time domain or area smoothing are used for this analysis? | Accepted | **We filtered the field in the spatial domain.**
the sentence about the filtering was revised:
**Pg4 lines 103-105 :**
'However, at high degrees and orders, GRACE spherical harmonics are contaminated by noise, including longitudinal stripes, and filtering is still needed. In our study, the smoothness priors method (Tarvainen et al., 2002; Zhan. et al., 2015) was used to remove noise in the spatial domain.'. |
| 7. Line 105: "the smoothness priors method (Tarvainen et al., 2002) was used to remove stripe noise in GRACE data", In CPCA analysis works best when dominant variability contained within the data set is concentrated in a narrow frequency band. In GRACE RL05 data has some spikes over equator region in some months. Authors should carefully handle the data, especially removing the spikes at the same time without losing the mass change signal. | Accepted | Before applying the CPCA, we have filtered each surface mass change field using the smoothness priors method and interpolated missing data with a spline function at each grid point.

We compared this filter with classical filters such as the Gaussian, Correlated-Error and combined filters (Gaussian with 300-km smoothing + Correlated-Error) in Zhan et al(2015). That work describes how the smoothness priors method works in removing noise in GRACE data, and compared the results of this filter with that of the Gaussian smoother, Correlated-Error filtering, and combined filter (Gaussian smoother + decorrelation filtering) with "actual signals". The results show that the smoothness priors method has the advantages of less reduction of signal amplitude at high latitude, retention of greater detail of short-wavelength components in the result, and less signal distortion at low latitude. Moreover, grid statistical results of the filtered field show that results of that method are the most similar to the actual minimum, maximum and RMS values of the original field. Please refer to Figures 1 and 2 and Table 1 listed below.

Figure 1a shows a numerical model simulation of mass |

change trend (as a 'true' signal), Figure 1b the simulation of a stripe noise model, and Figure 1c the synthesized signal of mass change trend from adding the data of Figure 1a and b. we then converted the field of Figure 1c into normalized spherical harmonic coefficients, to degree and order 60. Finally, we applied the smoothness priors method, Gaussian filter, correlated error filter, and combined filter to the synthesized signal.

Figure 2 shows the results of different filters. From these results, we can see that the outcome of the smoothness priors method (Figure 2d) has the advantages of less reduction in signal amplitude at high latitude, preservation of greater detail of short-wavelength components in the result, and less signal distortion at low latitude.

Table 1 lists grid statistical results of the numerical model of mass change (Figure 1a) and filtering results of mass change (Figure 2) from applying different filters in the synthetic mass change model. The grid statistical results of the filtered field show that the output of smoothness priors method is the most similar to the actual minimum, maximum and RMS values of the original field (Figure 1a).

We supplemented this information on **Pg4 line 105- Pg5 line 110** and Zhan et al (2015) on **P24 lines 602-603** in the revised manuscript. Thank you for the suggestions.

**Revised on Pg4 line 105- Pg5 line 110:**

'Compared with the Gaussian filter, Correlated-Error filter and the combined filter (Gaussian with 300 km smoothing + Correlated-Error), the smoothness priors method has advantages of less reduction in signal amplitude at high latitude, preservation of greater detail for short-wavelength components in the result and less signal distortion at low latitude. Moreover, grid statistical results of the filtered field show that the result of smoothness priors method is the most similar to the actual in the minimum, maximum and the RMS values of the original field (Zhan. et al.,2015)'.

**And the literature was supplemented:**

**Pg24 line 602-603:**

'Zhan, J.G., Wang Y., Shi H.L., Chai H., Zhu C.D.: Removing correlative errors in GRACE data by the smoothness priors method, Chinese J. Geophysics, 58(4):

1135-1144, 2015.'

[Figure]

Figure 1. (a) The numerical model of mass change trend, (b) the stripe noise model; (c) synthetic model by (a) + (b).

[Figure]

Figure 2. Results by applying different filters on the synthetic model. (a) The Gaussian filter with a smoothing radius of 300km; (b) the correlated error filter; (c) a 300km Gaussian smoothing after the correlated error filter; (d)the SPM filter.

Table 1.The grid statistics results of the numerical mass change trend model and the filtered mass change results by applying different filter on a synthetic mass change model. Unit: cm.

|  | minimum | maximun | mean | rms |  |
| --- | --- | --- | --- | --- | --- |
| original | -11.45 | 13.15 | -0.0396 | 1.448 |  |
| Gaussian | -6.78 | 11.06 | -0.0349 | 1.231 |  |
| De-correlation | -11.53 | 11.69 | -0.0399 | 1.727 |  |
| Combination | -6.53 | 11.10 | -0.0349 | 1.196 |  |
| SPM | -8.81 | 11.88 | -0.0399 | 1.371 |  |

| 8. line 145: "Wavelet amplitude...", authors did not explained all terms in the equation (9) and authors are not explained clearly the wavelet analysis methodology. The author should provide the recent literature regarding the wavelet analysis. | Accepted | We supplemented the information as "$C_\psi$ is a constant, and *a* and *b* are scale factors of period and time, respectively" and the recent literature in the revised manuscript. |
| --- | --- | --- |
| | | The "wavelet amplitude-period spectrum" was first presented by Professor L. Liu in his PhD thesis *Basic wavelet theory and its applications in geosciences*, and Zhan et al. (2003) applied this method to analyze the time-frequency character of sea level variation. Recently, Liu et al. (2012) presented "Inversion and Normalization of Time-Frequency Transform" and emphasized the inversion transform. It is very useful to extract the signal. |
| | | The wavelet amplitude-period spectrum can provide information on the instantaneous amplitude and period of each quasi-periodic signal, as well as phase information. We chose this method to analyze time-frequency characteristics of signals. |
| | | **The literature added to the revised manuscript on Pg7 lines 168-169:** |
| | | 'Here, we use the wavelet amplitude-period spectrum (**Liu, 1999; Liu and Hsu, 2012; Zhan et al., 2003**) to analyze…. '. |
| | | **Revised on Pg8 line 178:** |
| | | '$C_\psi$ is a constant, a and b are the scale factors of period and time, respectively'. |
| | | **Supplement:** |
| | | **Literature added to P21 lines 513-15 and P24 lines 600-601:** |
| | | 'Liu, L.: Basic wavelet theory and its applications in geosciences, Dissertation for the PHD of Solid Geophysics, institute of Geodesy and Geophysics, CAS, Wuhan, China, 1999. |
| | | Liu, L., and Hsu, H.: Inversion and Normalization of Time-Frequency Transform, Appl. Math, 6(1S): 67S-74S, 2012.' |
| | | 'Zhan, J.G., Wang Y., and L. Liu: Time-frequency analysis of the inter-seasonal variations of China-neighboring seas |

| | | level, Chinese J. Geophysics,46(1):36-41,2003.' |
|---|---|---|
| 9. Line159: "Figure 2 shows the trend of mass balance...", authors showed the spatial trends but did not mentioned how they calculated the trends? In Figure 2 is not clear and no trend scale. | Accepted | We supplemented the information in the revised manuscript and redrew Figure 2.

**Revised on Pg8 lines 181-186:**

'A regional $1° \times 1°$ gridded (24° - 45°N , 70°-105°E) surface mass change field (in units of equivalent water height) was calculated from each GRACE spherical harmonic solutions following Equations (1). Then, we filtered each surface mass change field using the smoothness priors method (Tarvainen et al., 2002; Zhan et al., 2015) and interpolated missing data using a spline function at each grid point. Finally, GRACE mass rate was estimated at each grid point using the least squares to fit a linear trend, plus annual and semiannual sinusoids to GRACE-derived mass change time series.';

Figure 2 Trend of mass balance in and around Tibetan Plateau. (A)Eastern Himalaya,(B)central Himalaya, (C)western Himalaya, (D)Pamirs, (E) Qiangtang Plateau,(F)Kunlun mountain,(G)Qinghai Plateau, (H) northwestern India. |
| 10. Generally CPCA analysis allows the efficient detection of propagating features, especially when the variance spread over a number of frequencies. Before applying the CPCA analysis the date shouldn't be discontinuous, sudden transitions, and noisy | Accepted | Before applying the CPCA, we filtered each surface mass change field using the smoothness priors method (Tarvainen et al., 2002) and interpolated missing data with a spline function at each grid point.

Thus, the data are continuous and there were no noisy spikes or sudden transitions before applying the CPCA.

We supplemented the information on **Pg8 line 190** in the revised manuscript:

**Pg8 line 190:**

'Before the CPCA analysis, data of mass change were |

| | | |
|---|---|---|
| spikes. | | filtered, and missing data were interpolated at each grid point.' |
| 11. In addition in this manuscript many conclusive statements, which are only speculations, and language corrections are required. | **Accepted** | Language corrections are also revised in the revised manuscript :

**Pg2 line 29:**

'In one hand, seawater absorbs' was changed to 'Seawater absorbs';

**Pg2 line 30:**

'On the other hand' was changed to 'Moreover ';

**Pg2 line 32:**

'also **raising…** ' was changed to 'which can also lead to rising …'.

**Pg2 line 35:**

'the Qinghai-Tibet Plateau' was changed to 'the Qinghai-Tibet Plateau (QTP)';

**Pg2 line 39:**

'Indian monsoon and East Asian monsoon ' was changed to 'Indian and East Asian monsoon ';

**Pg2 line 42:**

'The difference between their results is that Yi et al.' was changed to 'The difference between the results of those authors and that of Yi et al.(2014) is that the latter';

**Pg2 lines 43-44:**

'is so much stronger' was changed to 'is much stronger' ;

'and thus it can' was changed to 'and it can therefore' ;

'Pamirs precipitation' was changed to 'precipitation in the Pamirs';

**Pg3 line 56:**

'high' was changed to 'highly' ;

**Pg3 lines 60-61:**

'GRACE' was changed to 'Gravity Recovery and Climate Experiment (GRACE) observation data '; |

'signals from ' was changed to 'signals of';

**Pg3 line 62:**

'and the terrestrial water storage (TWS) model from the GRACE observation data, the residual gravity change can be totally attributed…' was changed to 'and terrestrial water storage model from GRACE data, residual gravity change can be fully attributed';

**Pg3 line 66:**

'time scales' was changed to 'temporal scales';

**Pg3 line 67:**

'The most advantage' was changed to 'The greatest advantage';

**Pg3 line 69:**

'standing wave' was changed to 'a standing wave';

**Pg4 line 82:**

'from the GRACE' was changed to 'from GRACE';

'have been' was changed to 'are';

**Pg4 line 83:**

'and its corresponding' was changed to 'and corresponding';

**Pg4 line 84:**

'by using' was changed to ' using '

**Pg4 line 85:**

'studied by ' was changed to ' examined' ;

**Pg4 line 86:**

'to explore the possible reasons' was changed to 'to explore possible reasons';

**Pg4 line 87:**

'… in the QTP, It is very helpful for us to understand the respond relationship ' was changed to '… over the QTP. This is very helpful to understand the response…';

**Pg4 line 88:**

'of great significance' was changed to 'very important' ;

| | | 'melting' was changed to 'melt'; |
| | | **Pg4 line 92:** |
| | | 'the America ' was changed to 'the U.S. ; |
| | | **Pg4 line 93:** |
| | | 'the changes' was changed to 'changes' ; |
| | | 'in 300-km' was changed to ' at 300-km'; |
| | | **Pg4 line 94:** |
| | | 'the change of hydrology and cryosphere' was changed to 'changes in hydrology and the cryosphere'; |
| | | **Pg5 line 113:** |
| | | 'Wahr et al. [1998],' was changed to 'Wahr et al. (1998)'. |
| | | 'equal ' was changed to 'equivalent '. |
| | | **Pg5 line 117:** |
| | | 'λ is colatitude, θ is latitude,' was changed to 'λ is longitude, θ is colatitude,'. |
| | | **Pg5 line 118:** |
| | | 'the normalized' was changed to ' normalized'; |
| | | **Pg6 line 139:** |
| | | 'Its complex form is' was changed to 'The constructed complex observation vector $U_j(t)$ can be expressed as'; |
| | | **Pg6 line 146:** |
| | | 'is the principal' was changed 'is principal'; |
| | | 'the CPCA' was changed to ' CPCA '; |
| | | **Pg6 line 147:** |
| | | 'the principal component' was changed to ' such'; |
| | | 'the complex vector constructed' was changed to 'the constructed complex vector'; |
| | | 'After the normalization' was changed to 'After normalization'; |
| | | **Pg7 line 167:** |
| | | 'After the temporal change series of principal components in |

the area being obtained' was changed to 'After obtaining the temporal change series of principal components in the area';

**Pg7 line 169:**

'choosing' was changed to ' choose ';

**Pg7 lines 168-169:**

'(Liu L., 1999; Liu L., and Hsu H., 2012; Zhan et al., 2003)' was added in the revised manuscript;;

**Pg8 line 178:**

' $C_\psi$ is a constant, a and *b* are scale factors of period and time' was added in the revised manuscript;

**Pg8 line 186:**

'period 2003 to 2015' was changed to 'period 2003-2015';

'From figure 2, we can see that mass balance' was changed to 'Figure 2 shows that this mass balance ';

**Pg8 line 188:**

'in' was changed to ' over';

**Pg8 line 190:**

'Before the CPCA analysis, data of mass change were filtered, and missing data were interpolated at each grid point' was added in the revised manuscript.

**Pg8 line 191:**

'shows the corresponding' was changed to 'shows corresponding'

**Pg8 line 192:**

'in this area' was changed as 'in the area';

**Pg8 line 194:**

'CPCA analysis of the mass variation in' was changed to 'CPCA of mass variation over the';

**Pg8 line 194-P9 line 195:**

'are respectively 82.65, 25.05 and 8.62, and their contribution percentages 54%, 16.37% and 5.64%,' was changed to ' are respectively 82.6516, 25.0562 and 8.6290, and their contribution percentages are respectively 54.02%,

| | | |
|---|---|---|
| | | 16.38% and 5.64%';

**Pg8 line 195:**

'could' was changed to 'can';

**Pg9 line 197:**

'is' was changed to' shows';

**Pg9 line 199:**

'eastern part of the Himalayas' was changed to 'eastern Himalayas';

**Pg9 line 200:**

'signal of the northwestern part of India' was changed to 'signal in the northwestern India';

**P9 line 203:**

'are the time evolution' was changed to ' depict the temporal evolution';

'**Pg9 line 204:**

'It can be seen in the figure 3c' was changed to 'It is seen in figure 3c';

'affect' was changed to 'affected';

'**Pg9 line 205:**

'its period' was changed to ' whose period ';

**Pg9 line 215:**

'mass variation' was changed to 'mass variation and the correlation analysis';

**Pg10 line 225:**

'oriented' was changed to 'orientation' ;

**Pg10 line 230:**

'are respectively the time evolution' was changed to 'respectively show the temporal evolution' ;

**Pg10 line 232:**

'we can see that ' was changed to 'we see that ';

**Pg10 line 234:** |

‘ of ’ was changed to ‘in’ ;

**Pg10 line 235:**

‘of the El Niño’ was changed to ‘of El Niño’ ;

**Pg10 line 241:**

‘the wavelet’ was changed to ‘ wavelet’;

‘we conclude’ was changed to ‘the data suggest’;

**Pg10 line 242:**

‘by East Asian’ was changed to ‘by climate change related to East Asian’;

**Pg10 line 243:**

‘one of the branches enters into the Qinghai plateau by through…’ was changed to ‘one enters the Qinghai Plateau through… ’ ;

**Pg10 line 244:**

‘ another branch enters ‘ was changed to ‘ and the other enters ’ ;

‘by through the eastern part of Himalayas ’ was changed to ‘ through the eastern Himalayas’ ;

**Pg10 line 245:**

‘until reaches’ was changed to ‘until reaching ‘;

‘turn to’ was changed to ‘ turns ‘ ;

**Pg10 line 246:**

‘From the figure, we can see’ was changed to ‘The figure shows … ’ ;

**Pg11 line 250:**

‘ has obvious character from west-to-east’ was changed to ‘had an obvious west-to-east configuration ’ ;

**Pg11 line 252:**

‘come’ was changed to ‘ came’ ;

**Pg11 line 252:**

‘Figure 5b and 5c show’ was changed to ‘Figure 5b and c shows’;

**Pg11 line 254:**

'In contrast with from…' was changed to 'In contrast with the results of ';

**Pg11 line 260:**

'moves to north ' was changed to 'moves to the north';

**Pg11 line 261:**

'enters Tarim ' was changed to 'enters the Tarim ';

'reaches to the eastern ' was changed to 'reaches the eastern ';

**Pg11 line 262:**

'moves to east beyond the west Himalayas and enters into the' was changed to 'moves east beyond the western Himalayas and enters the ';

**Pg11 line 263:**

'at 90°E area' was changed to 'around 90°E ';

**Pg11 line 267:**

'Mass Change of Mass in Inland Qinghai-Tibet Plateau' was changed to 'Mass Change in Inland QTP';

**Pg11 line 270:**

' Yi et al.(2014)' was changed to 'Yi and Sun (2014)' ;

**Pg11 line 274:**

'Jacob (2012) deduced the glacier' was changed to 'Jacob et al. (2012) deduced glacier';

**Pg12 line 276:**

'Qinghai-Tibet Plateau that area' was changed to ' that area' ;

'shows' was changed to ' have shown' ;

**Pg12 line 278:**

' 48.2m/ yr and the area was reduced' was changed to ' 48.2m yr$^{-1}$ and the area declined';

**P12 line 279:**

'0.57% /yr' was changed to '0.57% yr$^{-1}$';

'decreases from ' was changed to 'decreased from ' ;

**Pg12 line 284:**

'melted water' was changed to 'meltwater ';

**Pg12 line 286:**

'Yi et al. (2014)' was changed to 'Yi and Sun (2014) ';

**Pg12 line 291:**

'However, we believe that geologic structural processes are slow. Further, we still lack enough…'     was changed to 'However, we still lack enough…' ;

**Pg12 line 298:**

'believe' was changed to 'support the point ';

**Pg12 line 299:**

' on the ITP over the past…' was changed to ' over the inland QTP during the past…';

**Pg12 line 301:**

'influenced by El Niño ' was changed to 'On one hand, influenced by El Niño';

**Pg12 line 302- Pg13 line 303:**

'inland ' was changed to 'inland area' ;

**Pg13 line 306:**

'melting water' was changed to ' meltwater';

**Pg16 line 397:**

' Yi et al. (2014)' was changed to 'Yi and Sun (2014)';

**Pg16 line 398:**

'the correlation of mass' was changed to 'correlation between mass' ;

**Pg16 line 399:**

'Arctic Oscillation (AO), and found' was changed to 'the Arctic Oscillation (AO), founding';

**Pg16 line 400:**

'both the ENSO and AO ' was changed to 'both the ENSO and AO ' ;

**Pg17 line 412:**

'which is …' was changed to 'which was …' ;

'16.3%' was changed to '16.38%';

'Yi et al. (2014)' was changed to 'Yi and Sun (2014)';

**Pg17 line 414:**

'in QTP' was changed to 'in the QTP';

**Pg17 line 415:**

'phenomenon in the inland' was changed to ' on the inland' ;

**Pg17 line 420:**

'Conclusion' was changed to 'Conclusions';

**Pg17 line 421:**

'Mass change on' was changed to 'During 2003-2015, mass change on';

'varies' was changed to 'varied' ;

**Pg17 line 427:**

'74%' was changed to '76.04%';

**literature was supplemented:**

**Pg19 lines 450-451:**

[revised manuscript text omitted]

Zhan, J.G., Wang Y., Shi H.L., Chai H., Zhu C.D.: Removing correlative errors in GRACE data by the smoothness priors method, Chinese J. Geophysics, 58(4): 1135-1144, 2015. |

---

## Author Comment (AC3) · 1 Mar 2017

**Responses to the comments of referee 2**

| The comments from referee 2 | Responses | Changes in the manuscript |
|---|---|---|
| 1. Convincing evidences are required to support the authors' view. | Accepted | Some contents were added as the suggestion.

**Pg9 lines 210-214:**

[revised manuscript text omitted]

| 2. How was the mass balance obtained in Figure 2? | Accepted | We supplemented the relevant information in the revised manuscript and redrew Figure 2. |
| --- | --- | --- |

We supplemented the relevant information in the revised manuscript and redrew Figure 2.

**Revised on Pg8 line 180-185:**

'A regional 1° × 1° gridded (24° - 45°N , 70°-105°E) surface mass change field (in units of equivalent water height) was calculated from each GRACE spherical harmonic solutions following Equations (1). Then, we filtered each surface mass change field using the smoothness priors method (Tarvainen et al., 2002; Zhan et al., 2015) and interpolated missing data using a spline function at each grid point. Finally, GRACE mass rate was estimated at each grid point using the least squares to fit a linear trend, plus annual and semiannual sinusoids to GRACE-derived mass change time series'.

**The figure 2 was redrew and the trend scale was added.**

[Figure]

Figure 2 Trend of mass balance in and around Tibetan Plateau. (A)Eastern Himalaya,(B)central Himalaya, (C)western Himalaya, (D)Pamirs, (E) Qiangtang plateau,(F)Kunlun mountain,(G)Qinghai plateau, (H) northwestern India.

| 3. Why choose the smoothness priors method for filtering? The authors should explain more. | Accepted | Before applying the CPCA, we have filtered each surface mass change field using the smoothness priors method (Tarvainen et al.,2002) and interpolated missing data with a spline function at each grid point. |
|---|---|---|
| | | We compared this filter with classical filters such as the Gaussian, Correlated-Error and combined filters (Gaussian with 300-km smoothing + Correlated-Error) in Zhan et al(2015). That work describes how the smoothness priors method works in removing noise in GRACE data, and compared the results of this filter with that of the Gaussian smoother, Correlated-Error filtering, and combined filter (Gaussian smoother + decorrelation filtering) with "actual signals". The results show that the smoothness priors method has the advantages of less reduction of signal amplitude at high latitude, retention of greater detail of short-wavelength components in the result, and less signal distortion at low latitude. Moreover, grid statistical results of the filtered field show that results of that method are the most similar to the actual minimum, maximum and RMS values of the original field. Please refer to Figures 1 and 2 and Table 1 listed below. |
| | | Figure 1a shows a numerical model simulation of mass change trend (as a 'true' signal), Figure 1b the simulation of a stripe noise model, and Figure 1c the synthesized signal of mass change trend from adding the data of Figure 1a and b. we then converted the field of Figure 1c into normalized spherical harmonic coefficients, to degree and order 60. Finally, we applied the smoothness priors method, Gaussian filter, correlated error filter, and combined filter to the synthesized signal. |
| | | Figure 2 shows the results of different filters. From these results, we can see that the outcome of the smoothness priors method (Figure 2d) has the advantages of less reduction in signal amplitude at high latitude, preservation of greater detail of short-wavelength components in the result, and less signal distortion at low latitude. |
| | | Table 1 lists grid statistical results of the numerical model of mass change (Figure 1a) and filtering results of mass change (Figure 2) from applying different filters in the synthetic mass change model. The grid statistical results of the filtered field show that the output of smoothness priors method is the most similar to the actual minimum, maximum and RMS |

values of the original field (Figure 1a).

We supplemented this information on **P4 lines 105-Pg5 line110** and Zhan et al (2015) on **P22 lines 541-542** in the revised manuscript. Thank you for the suggestions.

**Revised on Pg4 line 105- Pg5 line110:**

'Compared with the Gaussian filter, Correlated-Error filter and the combined filter (Gaussian with 300 km smoothing + Correlated-Error), the smoothness priors method has advantages of less reduction in signal amplitude at high latitude, preservation of greater detail for short-wavelength components in the result and less signal distortion at low latitude. Moreover, grid statistical results of the filtered field show that the result of smoothness priors method is the most similar to the actual in the minimum, maximum and the RMS values of the original field (Zhan. et al.,2015).'

**And the literature was supplemented:**

**Pg24 line 601-602:**

'Zhan, J.G., Wang Y., Shi H.L., Chai H., Zhu C.D.: Removing correlative errors in GRACE data by the smoothness priors method, Chinese J. Geophysics, 58(4): 1135-1144, 2015.'

[Figure]

Figure 1. (a) The numerical model of mass change trend, (b) the stripe noise model; (c) synthetic model by (a) + (b).

[Figure]

Figure 2. Results by applying different filters on the synthetic model. (a) The Gaussian filter with a smoothing radius of 300km; (b) the correlated error filter; (c) a 300km Gaussian smoothing after the correlated error filter; (d)the SPM filter.

Table 1.The grid statistics results of the numerical mass change trend model and the filtered mass change results by applying different filter on a synthetic mass change model. Unit: cm.

|  | minimum | maximun | mean | rms |  |
|---|---|---|---|---|---|
| original | -11.45 | 13.15 | -0.0396 | 1.448 |  |
| Gaussian | -6.78 | 11.06 | -0.0349 | 1.231 |  |
| De-correlation | -11.53 | 11.69 | -0.0399 | 1.727 |  |
| Combination | -6.53 | 11.10 | -0.0349 | 1.196 |  |
| SPM | -8.81 | 11.88 | -0.0399 | 1.371 |  |

| 4. How to ensure reliable results with CPCA? | Accepted | Principal component analysis (PCA) was first formulated in statistics by Pearson (1901), who formulated the analysis to find "lines and planes of closest fit to systems of points in space". PCA was briefly mentioned by Fisher and MacKenzie (1923) as more suitable than analysis of variance for the modeling of response data. Fisher and MacKenzie also outlined the NIPALS algorithm, and Hotelling (1932) further developed PCA to its present state. Since then, the utility of PCA has been rediscovered in many diverse scientific fields, resulting in, amongst other things, an abundance of redundant terminology. PCA now goes under many names, such as singular value decomposition (SVD) (Golub et al., 1983; Mandel, 1982) and empirical orthogonal function analysis (Lagerloef et al., |

1988; Kaihatu et al., 1998; Zhang et al., 2004). Eigenvector and characteristic vector analysis are often used in the physical sciences. In image analysis, the term Hotelling transformation is often used for a principal component projection.

PCA (Abdi et al., 2010) is a multivariate technique that analyzes a data table, in which observations are described by several inter-correlated quantitative dependent variables. Its goal is to extract the important information from the table, represent it as a set of new orthogonal variables called principal components, and display the pattern of similarity of the observations and variables as points on maps. Mathematically, PCA depends upon the eigen-decomposition of positive semi-definite matrices and (SVD) of rectangular matrices.

Compared with PCA, the CPCA method (Horel, 1984) introduces phase information and has been shown to be a useful for identifying traveling and standing waves (Pfeffer et al., 2010; Kichikawa et al., 2015). CPCA transforms original data and its Hilbert transform into a complex time sequence and conducts PCA by calculating the covariance or complex characteristics vector of the cross-correlation matrix.

Thus, CPCA has well-developed theory and is a reliable method.

K. Pearson, On lines and planes of closest fit to systems of points in space, Philosophical Magazine, (6) 2 (1901)559-572.

R. Fisher and W. MacKenzie, Studies in crop variation. II. The manurial response of different potato varieties, Journal of Agricultural Science, 13 (1923) 311-320.

H. Hotelling, Analysis of a complex of statistical variables into principal components, Journal of Educational Psychology, 24 (1933) 417-441 and 498-520.

G. Golub and C. VanLoan, Matrix Computations, The Johns Hopkins University Press, Oxford, 1983

J. Mandel, Use of the singular value decomposition in regression analysis, American Statistician, 36 (1982) 15-24.

Lagerloef G S E, Bernstein R L. Empirical Orthogonal Function Analysis of Advanced Very High Resolution

Radiometer Surface Temperature Patterns in Santa Barbara Channel[J]. Journal of Geophysical Research, 1988, 93(93):6863-6873.

Kaihatu J M, Handler R A, Marmorino G O, et al. Empirical Orthogonal Function Analysis of Ocean Surface Currents Using Complex and Real-Vector Methods[J]. Journal of Atmospheric & Oceanic Technology, 1998, 15(4):927-941.

Zhang Y, Li T, Wang B. Decadal Change of the Spring Snow Depth over the Tibetan Plateau: The Associated Circulation and Influence on the East Asian Summer Monsoon. Journal of Climate, 2004, 17(14):2780-2793.

Abdi, H. and Williams, L. J. (2010), Principal component analysis. WIREs Comp Stat, 2: 433–459. doi:10.1002/wics.101

Horel J D. Complex Principal Component Analysis: Theory and Examples.[J]. Journal of Climatology & Applied Meteorology, 1984, 23(12):1660-1673.

Pfeffer R L, Ahlquist J, Kung R, et al. A study of baroclinic wave behavior over bottom topography using complex principal component analysis of experimental data[J]. Journal of the Atmospheric Sciences, 2010, 47(47):67-81.

Kichikawa Y, Arai Y, Iyetomi H. Complex Principle Component Analysis on Dynamic Correlation Structure in Price Index Data ☆[J]. Procedia Computer Science, 2015, 60(1):1836-1845.

We supplemented this information on **P5 line 120- Pg6 line134** in the revised manuscript.

**Revised on Pg5 line 121-Pg6 line135:**

'Principal component analysis (PCA) was first formulated in statistics by Pearson (1901), Hotelling (1932) further developed PCA to its present stage. Since then, the utility of PCA has been rediscovered in many diverse scientific fields, and it now goes under many names, such as singular value decomposition (SVD) (Golub et al., 1996; Mandel, 1982) and empirical orthogonal function (EOF) analysis (Lagerloef et al.,1988; Kaihatu et al.,1998; Zhang et al.,2004). Eigenvector analysis and characteristic vector analysis are

often used in the physical sciences and other fields.

PCA (Abdi et al.,2010; Helena et al., 2000; Wang et al.,2000) is a multivariate technique that analyzes a data table in which observations are described by several inter-correlated quantitative dependent variables. Its goal is to extract the important information from the table, represent it as a set of new orthogonal variables called principal components, and display patterns of similarity of the observations and variables as points in maps. Mathematically, PCA depends upon the eigen-decomposition of positive semi-definite matrices and SVD of rectangular matrices.

Compared with PCA, the CPCA method (Horel, 1984) introduces phase information and was shown to be a useful method for identifying traveling and standing waves (Pfeffer et al., 2010; Kichikawa et al., 2015). CPCA transforms original data and its Hilbert transform into a complex time sequence and conducts principal component analysis by calculating the covariance or complex characteristics vector of the cross-correlation matrix.'.

**literature was supplemented:**

**Pg19 line 450-451**

Abdi, H. and L. J. Williams: Principal component analysis, Wiley Interdisciplinary Reviews Computational Statistics, 2(4):433-459, 2010.

**Pg20 line 489-492:**

Golub G. and C. V. Loan: Matrix computations: John Hopkins University Press, 1996.

Helena, B., Pardo, R., Vega, M., Barrado, E., Fernandez, J. M., and L. Fernandez: Temporal evolution of groundwater composition in an alluvial aquifer (Pisuerga River, Spain) by principal component analysis, Water Research, 34(3):807-816, 2000.

**Pg20 line 495-496:**

Hotelling H.: Analysis of a complex of statistical variables into principal components, Journal of Educational Psychology, 24(6):417-520, 1932.

**Pg20 lines 499-501:**

Kaihatu, J. M., Handler, R. A., Marmorino, G. O., and L. K.

Shay: Empirical orthogonal function analysis of ocean surface currents using complex and real-vector methods, Journal of Atmospheric & Oceanic Technology, 15(4): 927-941, 1998.

**Pg21 line 504-509:**

Kichikawa Y, Arai Y. and H. Iyetomi: Complex Principle Component Analysis on Dynamic Correlation Structure in Price Index Data, Procedia Computer Science, 60(1):1836-1845, 2015.

Lagerloef G., R. L. Bernstein: Empirical Orthogonal Function Analysis of Advanced Very High Resolution Radiometer Surface Temperature Patterns in Santa Barbara Channel, Journal of Geophysical Research, 93(93):6863-6873, 1988.

Mandel J.: Use of the singular value decomposition in regression analysis, American Statistician, 36:15-24, 1982.

**Pg21 line 525- Pg22 line 528:**

Pearson K.: On lines and planes of closest fit to systems of points in space, Philosophical Magazine, (6) 2:559-572,1901.

Pfeffer, R. L., Ahlquist, J., Kung, R., Chang, Y., and G. Li: A study of baroclinic wave behavior over bottom topography using complex principal component analysis of experimental data, Journal of the Atmospheric Sciences, 47(47):67-81, 2010.

**Pg23 lines 570-571:**

Wang F. K. and T. Du: Using principal component analysis in process performance for multivariate data, Omega, 28(2):185-194, 2000.

**Pg24 line 597-598:**

Zhang Y, Li T. and B. Wang: Decadal Change of the Spring Snow Depth over the Tibetan Plateau: The Associated Circulation and Influence on the East Asian Summer Monsoon, Journal of Climate, 17(14):2780-2793, 2004.

| | | |
|---|---|---|
| 5. Is it suitable to only analyze the first three | Accepted | The reasons behind Tibetan Plateau glacier mass balance changes are very complicated. The CPCA showed its major components by calculating covariance of the cross- |

| | | |
|---|---|---|
| major factors? | | correlation matrix of mass balance. This paper only explains the first three major possible reasons for the mass change using the first three major components, which were responsible for 76.04% of that change. For more detailed information, one still needs to analyze the other principal components to explain the remaining 23.94% of mass balance. |
| 6. "However, we believe that geologic structural processes are slow." Could the authors quantitatively describe the impact on mass balance change? | Accepted | Here, we just want to express that geologic tectonics is a long process, and the period of 153 months is only a very short period relative to the geological tectonic time scale (millions of years). We still lack sufficient observation data of mass balance states in the interior part of the earth across the study region. We have removed the above sentence to avoid ambiguity on **P12 Lines 291-292** in the revised manuscript **Revised on Pg12 Lines 291-292:** 'However, we still lack enough observation data of mass balance states in the interior part of the earth in the study region.' |
| 7. The discussion and conclusions can be improved | Accepted | We have improved this content in the revised manuscript. Changes listed below: **Pg13 lines 303-305:** '
[revised manuscript text omitted]

|---|---|---|
| 8. There are no legends in some figures, such as Figure 2, Figure 3, | Accepted | We have added this information in the revised manuscript. |

Figure 4a and Figure 5a.

[Figure]

Figure 2 Trend of mass balance in and around Tibetan Plateau. (A)Eastern Himalaya,(B)central Himalaya, (C)western Himalaya, (D)Pamirs, (E) Qiangtang plateau,(F)Kunlun mountain,(G)Qinghai plateau, (H) northwestern India.

Figure 3 First spatial mode and phase (red arrows) (a), temporal patterns of first principal component (b), and its wavelet amplitude-period spectrum (c) of mass balance change, as well as wavelet amplitude-period spectrum of Indian monsoon indices in period 2003–2009 (d) .

Figure4 Second spatial mode and phase (red arrows) (a).

Figure 5 Third spatial mode and phase (red arrows) (a).

| 9. Line 22: the westerlies and of La Niña? | Accepted | We have revised this information on **Pg1 Line 23** in the revised manuscript. **Revised on Pg1 line 23:** It was changed to ' climate change from westerlies and La Niña'. |
|---|---|---|
| 10. Line 48: TBP should be explained when it first appears. | Accepted | We have revised this information on P2 Line35, and used the QTP instead of TBP and Qinghai-Tibet Plateau in the revised manuscript. **Revised on Pg2 line 35:** It was changed to 'QTP' |
| 11. Line 85: Qinghai Tibet Plateau? | Accepted | We have changed "Qinghai Tibet Plateau" to "QTP" on **Pg4 Line 87** in the revised manuscript. **Revised on Pg4 line 87:** It was changed to 'QTP'. |
| 12. Line 118: the left side of Eq. (3)? | Accepted | We have revised this information on **Pg6 Lines 139-140** in the revised manuscript, and the left side of Eq. (3) should show capital letter 'U'. **Revised on Pg6 lines 139-140:** 'The constructed complex observation vector $U_j(t)$ can be expressed as $U_j(t) = $'. |

| | Accepted | We have revised this information on **P8 Lines 194-195** in the revised manuscript. |
|---|---|---|
| 13. Line 168: The number in Table 1 should be same as the number in line 168 and in the paper. | | **Revised on Pg9 lines 194-195:**

'…respectively 82.6516, 25.0562 and 8.6290, and their contribution percentages are respectively 54.02%, 16.38% and 5.64%, which could explain 76.04% of the variation of…'. |
| 14. Line 273: the bracket? | Accepted | We have removed the brackets on **Pg13 lines 312-313** in the revised manuscript.

**Pg13 lines 312-313:**

'rate of -4.6 Gt yr$^{-1}$ in A area and -4.1 Gt yr$^{-1}$ in B area.' |
| 15. Line 310: 74%? | Accepted | It should be 76.04%.

We have revised this information **on P17 Line 427** in the revised manuscript.

**Revised on Pg17 line 427:**

'74%' was changed to ' 76.04% '. |
| | | Language corrections are also revised in the revised manuscript :

**Pg2 line 29:**

'In one hand, seawater absorbs' was changed to 'Seawater absorbs';

**Pg2 line 29:**

'In one hand, seawater absorbs' was changed to 'Seawater absorbs';

**Pg2 line 30:**

'On the other hand' was changed to 'Moreover ';

**Pg2 line 32:**

'also **raising…** ' was changed to 'which can also lead to rising …'.

**Pg2 line 35:** |

'the Qinghai-Tibet Plateau' was changed to 'the Qinghai-Tibet Plateau (QTP)';

**Pg2 line 39:**

'Indian monsoon and East Asian monsoon ' was changed to 'Indian and East Asian monsoon ';

**Pg2 line 42:**

'The difference between their results is that Yi et al.' was changed to 'The difference between the results of those authors and that of Yi et al.(2014) is that the latter';

**Pg2 lines 43-44:**

'is so much stronger' was changed to 'is much stronger' ;

'and thus it can' was changed to 'and it can therefore' ;

'Pamirs precipitation' was changed to 'precipitation in the Pamirs';

**Pg3 line 56:**

'high' was changed to 'highly' ;

**Pg3 lines 60-61:**

'GRACE' was changed to 'Gravity Recovery and Climate Experiment (GRACE) observation data ';

'signals from ' was changed to 'signals of';

**Pg3 line 62:**

'and the terrestrial water storage (TWS) model from the GRACE observation data, the residual gravity change can be totally attributed…'   was changed to   'and terrestrial water storage model from GRACE data, residual gravity change can be fully attributed';

**Pg3 line 66:**

'time scales' was changed to 'temporal scales';

**Pg3 line 67:**

'The most advantage' was changed to 'The greatest advantage';

**Pg3 line 69:**

'standing wave' was changed to 'a standing wave';

| | | **Pg4 line 82:** |
| | | 'from the GRACE' was changed to 'from GRACE'; |
| | | 'have been' was changed to 'are'; |
| | | **Pg4 line 83:** |
| | | 'and its corresponding' was changed to 'and corresponding'; |
| | | **Pg4 line 84:** |
| | | 'by using' was changed to ' using ' |
| | | **Pg4 line 85:** |
| | | 'studied by ' was changed to ' examined' ; |
| | | **Pg4 line 86:** |
| | | 'to explore the possible reasons' was changed to 'to explore possible reasons'; |
| | | **Pg4 line 87:** |
| | | '… in the QTP, It is very helpful for us to understand the respond relationship ' was changed to '… over the QTP. This is very helpful to understand the response…'; |
| | | **Pg4 line 88:** |
| | | 'of great significance' was changed to 'very important' ; |
| | | 'melting' was changed to 'melt'; |
| | | **Pg4 line 92:** |
| | | 'the America ' was changed to 'the U.S. ; |
| | | **Pg4 line 93:** |
| | | 'the changes' was changed to 'changes' ; |
| | | 'in 300-km' was changed to ' at 300-km'; |
| | | **Pg4 line 94:** |
| | | 'the change of hydrology and cryosphere' was changed to 'changes in hydrology and the cryosphere'; |
| | | **Pg5 line 113:** |
| | | 'Wahr et al. [1998],' was changed to 'Wahr et al. (1998)'. |
| | | 'equal ' was changed to 'equivalent '. |
| | | **Pg5 line 117:** |

'λ is colatitude, θ is latitude,' was changed to 'λ is longitude, θ is colatitude,'.

**Pg5 line 118:**

'the normalized' was changed to ' normalized';

**Pg6 line 139:**

'Its complex form is' was changed to 'The constructed complex observation vector $U_j(t)$ can be expressed as';

**Pg6 line 146:**

'is the principal' was changed 'is principal';

'the CPCA' was changed to ' CPCA ';

**Pg6 line 147:**

'the principal component' was changed to ' such';

'the complex vector constructed' was changed to 'the constructed complex vector';

'After the normalization' was changed to 'After normalization';

**Pg7 line 167:**

'After the temporal change series of principal components in the area being obtained' was changed to 'After obtaining the temporal change series of principal components in the area';

**Pg7 line 169:**

'choosing' was changed to ' choose ';

**Pg7 lines 168-169:**

'(Liu L., 1999; Liu L., and Hsu H., 2012; Zhan et al., 2003)' was added in the revised manuscript;;

**Pg8 line 178:**

' $C_\psi$ is a constant, a and *b* are scale factors of period and time' was added in the revised manuscript;

**Pg8 line 186:**

'period 2003 to 2015' was changed to 'period 2003-2015';

'From figure 2, we can see that mass balance' was changed

to 'Figure 2 shows that this mass balance ';

**Pg8 line 188:**

'in' was changed to ' over';

**Pg8 line 190:**

'Before the CPCA analysis, data of mass change were filtered, and missing data were interpolated at each grid point' was added in the revised manuscript.

**Pg8 line 191:**

'shows the corresponding' was changed to 'shows corresponding'

**Pg8 line 192:**

'in this area' was changed as 'in the area';

**Pg8 line 194:**

'CPCA analysis of the mass variation in' was changed to 'CPCA of mass variation over the';

**Pg8 line 194-P9 line 195:**

'are respectively 82.65, 25.05 and 8.62, and their contribution percentages 54%, 16.37% and 5.64%,' was changed to ' are respectively 82.6516, 25.0562 and 8.6290, and their contribution percentages are respectively 54.02%, 16.38% and 5.64%';

**Pg8 line 195:**

'could' was changed to 'can';

**Pg9 line 197:**

'is' was changed to' shows';

**Pg9 line 199:**

'eastern part of the Himalayas' was changed to 'eastern Himalayas';

**Pg9 line 200:**

'signal of the northwestern part of India' was changed to 'signal in the northwestern India';

**P9 line 203:**

'are the time evolution' was changed to ' depict the temporal

evolution';

**'Pg9 line 204:**

'It can be seen in the figure 3c' was changed to 'It is seen in figure 3c';

'affect' was changed to 'affected';

**'Pg9 line 205:**

'its period' was changed to ' whose period ';

**Pg9 line 215:**

'mass variation' was changed to 'mass variation and the correlation analysis';

**Pg10 line 225:**

'oriented' was changed to 'orientation' ;

**Pg10 line 230:**

'are respectively the time evolution' was changed to 'respectively show the temporal evolution' ;

**Pg10 line 232:**

'we can see that ' was changed to 'we see that ';

**Pg10 line 234:**

' of '   was changed to 'in' ;

**Pg10 line 235:**

'of the El Niño' was changed to 'of El Niño' ;

**Pg10 line 241:**

'the wavelet' was changed to ' wavelet';

'we conclude' was changed to 'the data suggest';

**Pg10 line 242:**

'by East Asian' was changed to 'by climate change related to East Asian';

**Pg10 line 243:**

'one of the branches enters into the Qinghai plateau by through…' was changed to 'one enters the Qinghai Plateau through… ';

| | | **Pg10 line 244:** |
| | | ' another branch enters ' was changed to ' and the other enters ' ; |
| | | 'by through the eastern part of Himalayas ' was changed to ' through the eastern Himalayas' ; |
| | | **Pg10 line 245:** |
| | | 'until reaches' was changed to 'until reaching '; |
| | | 'turn to' was changed to ' turns ' ; |
| | | **Pg10 line 246:** |
| | | 'From the figure, we can see' was changed to 'The figure shows … ' ; |
| | | **Pg11 line 250:** |
| | | ' has obvious character from west-to-east' was changed to 'had an obvious west-to-east configuration ' ; |
| | | **Pg11 line 252:** |
| | | 'come' was changed to ' came' ; |
| | | **Pg11 line 252:** |
| | | 'Figure 5b and 5c show' was changed to 'Figure 5b and c shows'; |
| | | **Pg11 line 254:** |
| | | 'In contrast with from…' was changed to 'In contrast with the results of '; |
| | | **Pg11 line 260:** |
| | | 'moves to north ' was changed to 'moves to the north'; |
| | | **Pg11 line 261:** |
| | | 'enters Tarim ' was changed to 'enters the Tarim '; |
| | | 'reaches to the eastern ' was changed to 'reaches the eastern '; |
| | | **Pg11 line 262:** |
| | | 'moves to east beyond the west Himalayas and enters into the' was changed to 'moves east beyond the western Himalayas and enters the '; |

**Pg11 line 263:**

'at 90°E area' was changed to 'around 90°E ';

**Pg11 line 267:**

'Mass Change of Mass in Inland Qinghai-Tibet Plateau' was changed to 'Mass Change in Inland QTP';

**Pg11 line 270:**

' Yi et al.(2014)' was changed to 'Yi and Sun (2014)' ;

**Pg11 line 274:**

'Jacob (2012) deduced the glacier' was changed to 'Jacob et al. (2012) deduced glacier';

**Pg12 line 276:**

'Qinghai-Tibet Plateau that area' was changed to ' that area' ;

'shows' was changed to ' have shown' ;

**Pg12 line 278:**

' 48.2m/ yr and the area was reduced' was changed to ' 48.2m yr$^{-1}$ and the area declined';

**P12 line 279:**

'0.57% /yr' was changed to '0.57% yr$^{-1}$';

'decreases from ' was changed to 'decreased from' ;

**Pg12 line 284:**

'melted water' was changed to 'meltwater ';

**Pg12 line 286:**

'Yi et al. (2014)' was changed to 'Yi and Sun (2014) ';

**Pg12 line 291:**

'However, we believe that geologic structural processes are slow. Further, we still lack enough…'    was changed to 'However, we still lack enough…' ;

**Pg12 line 298:**

'believe' was changed to 'support the point ';

**Pg12 line 299:**

' on the ITP over the past…' was changed to ' over the inland

QTP during the past…';

**Pg12 line 301:**

'influenced by El Niño ' was changed to 'On one hand, influenced by El Niño';

**Pg12 line 302- Pg13 line 303:**

'inland ' was changed to 'inland area' ;

**Pg13 line 306:**

'melting water' was changed to ' meltwater';

**Pg16 line 397:**

' Yi et al. (2014)' was changed to 'Yi and Sun (2014)';

**Pg16 line 398:**

'the correlation of mass' was changed to 'correlation between mass' ;

**Pg16 line 399:**

'Arctic Oscillation (AO), and found' was changed to 'the Arctic Oscillation (AO), founding';

**Pg16 line 400:**

'both the ENSO and AO ' was changed to 'both the ENSO and AO ' ;

**Pg17 line 412:**

'which is …' was changed to 'which was …' ;

'16.3%' was changed to '16.38%';

'Yi et al. (2014)' was changed to 'Yi and Sun (2014)';

**Pg17 line 414:**

'in QTP' was changed to 'in the QTP';

**Pg17 line 415:**

'phenomenon in the inland' was changed to ' on the inland' ;

**Pg17 line 420:**

'Conclusion' was changed to 'Conclusions';

**Pg17 line 421:**

'Mass change on' was changed to 'During 2003-2015, mass

change on';

'varies' was changed to 'varied' ;

**Pg17 line 427:**

'74%' was changed to '76.04%';

**literature was supplemented:**

**Pg19 lines 450-451:**

[revised manuscript text omitted]

---

## Referee Report (RR1)

TC-2016-259R

Review on "Complex Principal Component Analysis of Mass Balance change on Qinghai-Tibet Plateau" by Jingang Zhan et al.,

This paper diagnosed on GRACE mass balance wave characteristics using CPCA method during from January 2003 to September 2015. I appreciate the response from the authors. The authors addressed most of the concerns raised in the first review. But still, some points need to be clarified. I urge the authors carefully check the manuscript before it can be accepted for publication.

In the response to the reviewer comments, the authors specifically mention that "we supplemented the information in the revised manuscript. GRACE mass rate was estimated at each grid point using least squares to fit a linear trend, plus annual and semi-annual..". Regarding the trend analysis, the reviewer had asked the methodology and how they calculated. The calculated trend values are with or without annual and semi-annual trends and authors should clearly mention in the manuscript.

Figure 2 shows the regional 1x1 gridded data of GRACE used. But GRACE is providing 3 x 3 grids and also lot of spikes especially in the equatorial region. Authors should mention the accuracy of the GRACE data, especially 1x1 grids.

Legends, x, and y-axis labels are not clear in all figure.

---

## Author Response (AR2)

Dear editor:

Thanks very much for you and Dr. Kishore for the helpful advices and comments that will help us to improve our manuscript.

The comments were carefully considered. Some contents were added in the manuscript to dispel the concerns and make it easier for readers to understand based on the suggestions of referee, the relevant references were also added in the revised manuscript. Legends and labels were also adjusted to make them clearer in the revised manuscript. All changes in revised manuscript were marked in red.

**Some contents were added in the revised manuscript to make it easier for readers to understand based on the suggestion.**

**Pg4 lines 102-103:**

'and has the ability to monitor 1 mm geoid undulation at the spatial scale of 300 km ( Bettadpur et al,2015; Save et al.,2016 ).'

**Pg8 lines 186-187:**

'As the fitting results, the amplitude values of annual and semiannual terms are constants, so the calculated trend values contain the contributions from the annual and semiannual trend.'

**Pg8 lines 187-196:**

'It's worth noting that, $1° \times 1°$ gridded data used here does not represent that the resolution of GRACE data has been improved. The resolution of the calculated data depends on the degree of the RL05 solutions and the GRACE RL05 solutions are limited by the band-limited nature of GRACE orbit configuration (inclination, altitude, and separation of the twin satellites), with an approximate resolution of around 300 km near the equator (Chen et al., 2016). People can also get relevant information from NASA websites (https://grace.jpl.nasa.gov/data/get-data/jpl_global_mascons/). One can also calculate smaller grid data using those solutions, the smaller calculated grid data does not mean that there are more short-wavelength signals in the results and the accuracy of the calculated data is still 1 mm geoid undulation at around 300 km scale. The accuracy of the calculated grid data depends on the accuracy of the RL05 solution itself (Bettadpur et al,2015; Save et al.,2016), not the size of the grid.'

**literature was supplemented:**

**Pg19 lines 469-472:**

'Bettadpur S, Kang Z., Peter N., Rick P., Steve P., John R. and H. Save: Status and Assessments of CSR GRACE Level-2 Data Products,EGU General Assembly Conference, 2015. [ Available at http://www.gfz-potsdam.de/en /section/global-geomonitoring-and-gravity-field /topics/ development-operation-and-analysis-of-gravity-field-satellite-missions/grace /gstm /gstm-2014/proceedings/ ].'

**Pg20 lines 485-486:**

'Chen, J. L., C. R. Wilson, B. D. Tapley, H. Save, and J.-F. Cretaux: Longterm and seasonal Caspian Sea level change from satellite gravity and altimeter measurements, J. Geophys. Res. Solid Earth, 122, 2274–2290, 2017.'

**Pg22 lines 553-554:**

Save H, Bettadpur S., B. D. Tapley: High-resolution CSR GRACE RL05 mascons, Journal of Geophysical Research Solid Earth, 121(10):7547–7569, 2016.

**Responses to the comments of Dr. Kishore**

2017-5-4

| The comments from referee 1 | Responses | Changes made in the manuscript |
|---|---|---|
| 1. In the response to the reviewer comments, the authors specifically mention that "we supplemented the information in the revised manuscript. GRACE mass rate was estimated at each grid point using least squares to fit a linear trend, plus annual and semi-annual..". Regarding the trend analysis, the reviewer had | Accepted | For the time series of mass change, we used the following model to fit the linear trend. $$y(t) = c + bt + A_1 \cos(2\pi t / P_1 + \varphi_1) + A_2 \cos(2\pi t / P_2 + \varphi_2)$$ As the fitting results, the amplitude values of annual and semiannual terms are constants, so the calculated trend values contain the contributions from the annual and semiannual trend.

 We supplemented this information on Pg8 lines 186-187 as the suggestion. Thank you for the suggestions.
**Revised on Pg8 lines 186-187:**
 'As the fitting results, the amplitude values of annual and semiannual terms are constants, so the calculated trend values contain the contributions from the annual and semiannual trend.' |

| | | |
|---|---|---|
| asked the methodology and how they calculated. The calculated trend values are with or without annual and semi-annual trends and authors should clearly mention in the manuscript. | | |
| 2. Figure 2 shows the regional 1x1 gridded data of GRACE used. But GRACE is providing 3 x 3 grids and also lot of spikes especially in the equatorial region. Authors should mention the accuracy of the GRACE data, especially 1x1 grids. | Accepted | The time-varying gravity model used in this paper is the Release-05 (RL05) solutions provided by the Center for Space Research (CSR), University of Texas at Austin. Each of them consists of normalized spherical harmonic (SH) coefficients, to degree and order 60. The RL05 production already has the ability to monitor 1 mm geoid undulation at the spatial scale of 300 km (Bettadpur et al,2015; Save et al.,2016). Please refer to the following figure(from Bettadpur,2015). The figure describes that the Geoid height RMS for RL05 data is no more than 1mm.

It's worth noting that, $1° × 1°$ gridded data here does not represent that the resolution of GRACE data has been improved. The resolution of the calculated data depends on the degree of the RL05 solutions and the GRACE RL05 solutions are limited by the band-limited nature of GRACE orbit configuration (inclination, altitude, and separation of the twin satellites), with an approximate resolution of around 300 km near the equator (Chen et al., 2016). People can also get relevant information from NASA websites (https://grace.jpl.nasa.gov/data/get-data/jpl_global_mascons/). One can also calculate smaller grid data using those solutions, the smaller calculated grid data does not mean that there are more short-wavelength signals in the results and the accuracy of the calculated data is still 1 mm geoid undulation at around 300 km scale. The accuracy of the calculated grid data depends on the accuracy of the RL05 solution itself (Bettadpur et al,2015; Save et al.,2016), not the size of the grid. |

[Figure]

[Figure]

We supplemented this information on Pg4 lines 102-103 and P8 lines 187-196, the references (Bettadpur et al 2015; Save et al.,2016;Chen et al.,2016) are also supplemented on P19 lines 469-472 , P20 lines 485-486 and P22 lines 553-554 in the revised manuscript as the suggestion. Thank you for the suggestions.

**Pg4 lines 102-103:**

'and has the ability to monitor 1 mm geoid undulation at the spatial scale of 300 km (Bettadpur et al,2015; Save et al., 2016 ).'

**P8 lines 187-196:**

'It's worth noting that, $1° \times 1°$ gridded data used here does not represent that the resolution of GRACE data has been improved. The resolution of the calculated data depends on the degree of the RL05 solutions and the GRACE RL05 solutions are limited by the band-limited nature of GRACE orbit configuration (inclination, altitude, and separation of the twin satellites), with an approximate resolution of around 300 km near the equator (Chen et al., 2016). People can also get relevant information from NASA websites (https://grace.jpl.nasa.gov/data/get-data/jpl_global_mascons/). One can also calculate smaller grid data using those solutions, the smaller calculated grid data does not mean that there are more short-wavelength signals in the results and the accuracy of the calculated data is still 1 mm geoid undulation at around 300 km scale. The accuracy of the calculated grid data depends on the accuracy of the RL05 solution itself (Bettadpur et al,2015; Save et al.,2016), not the size of the grid.'

**Pg19 lines 469-472:**

'Bettadpur S, Kang Z., Peter N., Rick P., Steve P., John R. and H. Save: Status and Assessments of CSR GRACE Level-2 Data Products,EGU General Assembly Conference, 2015. [Available at http://www.gfz-potsdam.de/en /section/global-geomonitoring-and-gravity-field /topics/ development-operation-and-analysis-of-gravity-field-satellite-missions/grace /gstm /gstm-2014/proceedings/].'

**Pg20 lines485-486:**

'Chen, J. L., C. R. Wilson, B. D. Tapley, H. Save, and J.-F. Cretaux: Longterm

| | | and seasonal Caspian Sea level change from satellite gravity and altimeter measurements, J. Geophys. Res. Solid Earth, 122, 2274–2290, 2017.'

**Pg22 lines 553-554:**

'Save H., Bettadpur S, B. D. Tapley: High-resolution CSR GRACE RL05 mascons, Journal of Geophysical Research Solid Earth, 121(10):7547–7569, 2016.' |
|---|---|---|
| 3. Legends, x, and y-axis labels are not clear in all figure。 | Accepted | We adjusted the Legends and labels to make them clearer in the revised manuscript. |
| | | |

[revised manuscript text omitted]